# MoRGen: Mixture-of-Resolutions Generative Forecasting for Irregularly Sampled Medical Time-Series Data

Nassim Oufattole [1]  Matthew McDermott [2]  Collin Stultz [1 3]

## Abstract

Autoregressive generative models for irregularly sampled clinical time-series data are increasingly used for zero-shot risk forecasting. Prior work typically adopts a single fine-grained discretization of time, where tokens are generated at one fixed, pre-determined temporal resolution. We demonstrate that the zero-shot accuracy of individual generative forecasters varies with temporal resolution: performance can degrade when the model resolution is poorly matched to the temporal dynamics of the endpoint being evaluated. We then propose MoRGen (Mixture-of-Resolutions Generation), which fuses forecasts from generative experts trained at multiple temporal resolutions using a low-capacity task-specific mixture, improving performance across tasks with different temporal dynamics. Across multiple horizons and outcomes on three independent clinical datasets, MoRGen achieves lower binary cross-entropy (BCE) and statistically significant AUROC gains over autoregressive generative models that forecast tokens at a fixed temporal resolution.

## 1. Introduction

Risk stratification is an important component of clinical care. Identifying those patients who are at the highest risk of adverse outcome helps to ensure the right therapy is matched to the patient, based on their risk of future adverse events. For example, aggressive therapies in low risk patients are often associated with significant risks that outweigh any potential benefit. Hence, it is important to ensure that aggressive therapies are used in patients at the highest risk of adverse outcomes. Effective risk stratification also enables patient risk-informed post-discharge planning to reduce acute care re-utilization (Jack et al., 2009; Hernandez et al., 2010).

[1]MIT [2]Columbia University [3]Harvard Medical School. Correspondence to: Nassim Oufattole <nassim@mit.edu>.

*Proceedings of the 43rd International Conference on Machine Learning*, Seoul, South Korea. PMLR 306, 2026. Copyright 2026 by the author(s).

Generative models form an attractive platform for predicting the risk of future events. As the models themselves predict future tokens in an autoregressive manner, they can be used to determine the risk of a variety of different clinical endpoints. Recent work suggests that zero-shot generative models have the capability to achieve high AUROC on various clinical tasks (Renc et al., 2024; 2025). Modeling patient trajectories, however, is challenging due to noise, missingness, and the inherent challenges associated with modeling sparse irregularly sampled time series data (Tayefi et al., 2021; Xiao et al., 2018).

Another challenge associated with obtaining good zero-shot performance is that generative models typically produce trajectories at a fixed temporal resolution; e.g., one token per minute or day. Important clinical outcomes, however, may have very different temporal dynamics. For example, a serum potassium level can change quickly - on the order of minutes to hours, while other clinical measurements can take weeks to change (e.g., the hemoglobin A1c). Case in point, suppose our task is to predict a given outcome, k, within a pre-specified time interval, $[0, H_k]$, where $H_k$ is the horizon for the prediction task; e.g., predicting death, 1 year after being discharged for heart failure ($H_k = 1$ year). A model designed to generate tokens at a resolution as low as a minute, must produce potentially extremely long sequences to represent a one-year horizon. This creates a fundamental mismatch between the granularity of the generative process and the timescale of the clinical endpoint: long-horizon forecasting becomes mediated by many intermediate generation steps whose fine-scale temporal detail may not contribute significantly to the accuracy of estimating an endpoint probability. An alternative is to change the learning problem: instead of attempting to reconstruct the entire high-resolution future, we can forecast at coarser temporal resolutions (e.g., month-level) and model whether clinically meaningful events occur within each (month-sized) window. Coarsening trades temporal detail for stability and can emphasize different signals. In this paper, we empirically demonstrate across a range of datasets, tasks, and horizons that the optimal resolution is not one-size-fits-all: varying temporal resolution induces systematic tradeoffs in discrimination, calibration, and sharpness.

*Table 1.* Capabilities and temporal abstractions in prior EHR sequence models.

| | Event-stream | Coarse windows | Forecast eval | Time-gaps | Vary $\Delta$ | Fusion |
|---|---|---|---|---|---|---|
| ETHOS (Renc et al., 2024) | ✓ | | ✓ | ✓ | | |
| ARES (Renc et al., 2025) | ✓ | | ✓ | ✓ | | |
| Event Stream GPT (McDermott et al.) | ✓ | | ✓ | ✓ | | |
| TransformEHR (Yang et al., 2023) | | ✓ | ✓ | ✓ | | |
| CEHR-BERT (Pang et al., 2021) | | ✓ | ✓ | ✓ | | |
| Hi-BEHRT (Li et al., 2023) | | ✓ | ✓ | ✓ | | |
| Foresight (Kraljevic et al., 2024) | ✓ | | ✓ | | | |
| HALO (Theodorou et al., 2023) | | ✓ | | | | |
| EHR-GPT (Redekop et al., 2025) | | ✓ | ✓ | ✓ | | |
| MoRGen (Ours) | ✓ | ✓ | ✓ | ✓ | ✓ | ✓ |

To capitalize on these tradeoffs, we propose Mixture of Resolutions Generative forecasting (MoRGen), which fuses multiple resolution-specific generative experts to exploit these complementary strengths across generative model resolutions. This achieves superior AUROCs across a wide array of endpoints while adding minimal additional inference latency–since forecasting with coarser experts is computationally cheap relative to the fine-grained baseline. Additionally, by adhering to the MEDS (McDermott et al., 2025) standard, we maintained identical and reproducible pipelines across the datasets used in this study, while facilitating the application of MoRGen to other MEDS-compatible datasets with minimal engineering effort.[1]

## 2. Related Work

Table 1 summarizes how prior EHR sequence models choose a temporal abstraction and what they evaluate. **Event-stream** indicates tokens correspond to timestamped clinical events (labs, codes, etc.) in a fine-grained sequence. **Coarse windows** indicates tokens group events into larger units, often in encounter/visit containers (variable-length windows) in related past works, which implicitly coarsen time. **Forecast eval** indicates explicit evaluation on forecasting metrics (e.g., AUROC). **Time-gaps** indicates explicit time-gap / time-stamp tokens or embeddings are used in the sequence model. **Vary $\Delta$** indicates an explicit comparison over different window sizes (e.g., day/week/month), rather than relying on a single resolution for modeling. **Fusion** indicates combining multiple temporal abstractions/resolutions in a single predictor.

**Fine-grained event streams.** A first line of work models patient records as fine-grained event streams, where each token corresponds to a clinical event and irregular sampling is handled explicitly (e.g., via time-gap tokens). ETHOS (Renc et al., 2024) and AReS (Renc et al., 2025) show

that such autoregressive event-stream models can support *zero-shot* risk forecasting by rolling out futures and estimating horizon-specific event probabilities from sampled continuations. Event Stream GPT (McDermott et al.) is closely related in spirit, and Foresight (Kraljevic et al., 2024) also follows an event-stream style representation, but these works do not set out to study how *fixed* temporal coarsening (day/week/month) changes forecast quality across tasks, nor how combining resolutions can improve forecast performance.

**Encounter/visit-level windows and other objectives.** A second family uses coarser, encounter/visit-level containers, where each visit forms a set or short sequence of codes and the patient record becomes a sequence of visits. This representation implicitly coarsens time using variable-time-length windows defined by clinical encounters, and it is often paired with time features (e.g., time-since-last-visit or visit-time embeddings), as in CEHR-BERT (Pang et al., 2021), Hi-BEHRT (Li et al., 2023), and TransformEHR (Yang et al., 2023). We also note some generative models emphasize synthetic record generation rather than forecast evaluation (e.g., HALO (Theodorou et al., 2023)), while others target zero-shot forecasting but still rely on encounter-defined windows (e.g., EHR-GPT (Redekop et al., 2025)). Across these approaches, the "window" is typically dataset-/system-defined (encounter boundaries, visit timestamps) rather than a tunable, fixed-window time length, $\Delta$, chosen to match outcome dynamics.

**Our contribution: fixed-resolution sweeps and fusion.** In contrast to prior work that commits to either fine-grained event streams or encounter-defined coarsening, we *explicitly sweep fixed temporal resolutions* $\Delta$ (e.g., day/week/month) and show that no single resolution dominates across outcomes and horizons. We then fuse across resolutions to exploit complementary, resolution-dependent errors, improving discriminative performance. Conveniently for development, our coarsening does not require dataset-specific visit codes or encounter definitions: it is controlled by a fixed window size, making the resolution choice explicit, comparable across datasets, and directly analyzable.

## 3. Problem Setup and Evaluation

**Tasks and horizons.** We consider a set of binary forecasting tasks indexed by $k \in \{1, \ldots, K_{\text{tasks}}\}$[2] and define the binary outcome:

$$Y_j^{(k)} \in \{0, 1\},$$

where $Y_j^{(k)} = 1$ indicates that outcome $k$ occurs at least once within a pre-specified interval $(0, H_k]$ (where $H_k$ is the horizon for task k), and $j$ is the patient index. If $k$ does

---

[1]Code available at: https://github.com/Oufattole/morgen

[2]Note we suppress the $k$ in notation outside this section.

not occur once within $(0, H_k]$, then $Y_j^{(k)} = 0$.

The *true* patient-specific risk is

$$\pi_j^{(k)} \ := \ \mathbb{P}\Big(Y_j^{(k)} = 1 \mid h_j\Big).$$

where $h_j$ denotes the patient's prior history (including diagnoses, procedures, medications, and laboratory measurements), and corresponds to an irregularly sampled event sequence.

**Zero-shot generative forecasting.** A trained generative model with parameters $\theta$ induces a predictive distribution over future trajectories conditioned on history $h_j$. For each task $k$ the model implies the horizon event probability

$$\hat{\pi}_j^{(k)}(\theta) \ := \ \mathbb{P}_\theta\Big(Y_j^{(k)} = 1 \mid h_j\Big).$$

For these studies we focus on forecasting patient trajectories after being discharged from a hospital admission.

In practice, we estimate $\hat{\pi}_j^{(k)}(\theta)$ via Monte Carlo rollouts: sample $S$ future trajectories from the model conditioned on $h_j$ and compute the fraction of sampled futures in which the event occurs within horizon $H_k$,

$$\hat{\pi}_j^{(k)} \ := \ \frac{1}{S} \sum_{s=1}^{S} \mathbf{1}\{\text{outcome } k \text{ occurs within } H_k \text{ in rollout } s\}.$$

When $S$ is large, $\hat{\pi}_{j,H}^{(k)}$ is a low-variance estimator of the model-implied probability $\pi_j^{(k)}(\theta)$.

**Temporal resolution.** A key design choice is the temporal resolution $\Delta$ used to represent and generate future trajectories. We compare models trained at different resolutions, and write $\hat{\pi}_j^{(k)}(\theta, \Delta)$ for the horizon risk estimate produced by a model operating at resolution $\Delta$.[3]

Our central question is: *how does forecast quality vary with $\Delta$, and can we combine multiple resolutions to improve performance across heterogeneous outcomes and horizons?*

### 3.1. Evaluation metrics

We evaluate probabilistic forecasts along three complementary axes: **discrimination**, **calibration**, and **sharpness**. We additionally report **binary cross-entropy (BCE)**.

**Discrimination (AUROC).** We report the area under the ROC curve (AUROC), computed from predicted risks $\hat{\pi}_j^{(k)}$ and labels $Y_j^{(k)}$ for each task and horizon. AUROC measures a model's ability to rank patients by risk and is

---

[3]Note we suppress the $\theta$ and $\Delta$ in notation.

threshold-free, making it standard for clinical risk stratification (Renc et al., 2024; 2025; Jeong et al., 2024; Wornow et al., 2025; Steinberg et al., 2021).

**Overall probabilistic accuracy (BCE).** We report the binary cross-entropy (log loss) as well:

$$\text{BCE} := \frac{1}{N} \sum_{j=1}^{N} \Big[ -Y_j^{(k)} \log \hat{\pi}_j^{(k)} - (1 - Y_j^{(k)}) \log(1 - \hat{\pi}_j^{(k)}) \Big].$$

**Calibration (binned calibration error).** To quantify how well predicted probabilities match empirical event rates, we estimate a reliability-style calibration error using quantile bins of $\hat{\pi}_j^{(k)}$. Let $b(j) \in \{1, \ldots, B\}$ denote the bin assignment, with empirical bin mass $\hat{w}_b$. Define the binwise empirical event rate and mean prediction

$$\tilde{q}_b := \mathbb{E}[Y \mid b], \qquad \bar{\pi}_b := \mathbb{E}[\hat{\pi} \mid b],$$

estimated by sample means on a held-out split. We then compute a KL-based calibration error (called reliability in related works (Weijs et al., 2010; Benedetti, 2010; Murphy & Winkler, 1977; Bröcker, 2009))

$$\text{CalErr} := \sum_{b=1}^{B} \hat{w}_b \, D_{\text{KL}}\Big(\text{Ber}(\tilde{q}_b) \,\big\|\, \text{Ber}(\bar{\pi}_b)\Big).$$

Lower values indicate better calibration.

**Sharpness (confidence proxy).** Calibration does not capture how decisive a model's probabilistic forecasts are. We therefore report a simple sharpness proxy based on the average Bernoulli variance of the predicted probabilities:

$$\text{Var}(\hat{\pi}) \ := \ \mathbb{E}_j \Big[ \hat{\pi}_j^{(k)} \big( 1 - \hat{\pi}_j^{(k)} \big) \Big].$$

Lower variance values correspond to more concentrated predictions (more mass near 0 or 1)–higher sharpness. Higher values indicate less decisive forecasts (mass nearer 0.5)–lower sharpness. This quantity is label-free and is used only for *relative* comparisons across models; sharpness is not inherently better without calibration, since a model can be sharp but miscalibrated (overconfident). We therefore interpret sharpness jointly with Calibration Error and BCE in Section 5.

## 4. Single- and Multi-Resolution Generative Forecasting

This section introduces (i) our single-resolution baseline generative forecaster, (ii) coarse-resolution models obtained by fixed-window tokenization and Bernoulli-mixture compression, (iii) a simple mixture-of-resolutions fusion method that exploits complementary strengths across resolutions.

### 4.1. Single-Resolution Baseline: Event-Time and Observation Tokens

Our baseline follows prior autoregressive EHR sequence models in representing a patient trajectory as a sequence of discrete tokens describing *when* events occur and *what* is observed. Given history $h_j$, the model generates a post-discharge continuation by alternating between: (i) a *time-gap token* encoding the elapsed time since the previous event and (ii) one or more *observation tokens* encoding the clinical events at that time.

**Vocabulary.** The vocabulary contains: (i) discrete time-gap tokens, with a minimum granularity of one minute, and (ii) discrete observation tokens for structured clinical events (codes) and numeric measurements. Numeric values are discretized by deciles (computed on the training set) and treated as categorical tokens; e.g., a laboratory measurement contributes a token identifying the lab plus a token for its decile bin.

**Autoregressive modeling.** Let $x_{j,1:T}$ denote the tokenized post-discharge trajectory at the baseline resolution. We train a decoder-only autoregressive model $p_\theta$ with next-token log likelihood,

$$p_\theta(x_{j,1:T}) = \prod_{t=1}^{T} p_\theta(x_{j,t} \mid x_{j,<t}).$$

$$\min_\theta \ -\sum_j \sum_{t=1}^{T} \log p_\theta(x_{j,t} \mid x_{j,<t}).$$

At inference, we condition on the tokens in $h_j$ and sample $S = 50$ rollouts and estimate horizon probabilities $\hat{\pi}_j$ for each outcome and horizon as described in Section 3.

### 4.2. Coarse Resolution via Window Tokenization and Bernoulli-Mixture Compression

A limitation of the baseline is that long-horizon forecasting requires generating long token sequences, since time gaps are resolved down to minutes. To study the effect of temporal granularity and to build more stable long-horizon forecasters, we construct coarse-resolution models parameterized by a window size $\Delta$ (e.g., day, week, month).

**Windowed presence representation.** Partition the post-discharge timeline into contiguous windows of length $\Delta$. For each window, we form a binary *presence vector* $v \in \{0,1\}^{|\mathcal{V}|}$ indicating which vocabulary items appear at least once within the window.[4]

---

[4] For numeric measurements, the vocabulary includes discretized bins; presence indicates whether any value fell into that bin in the window. Variants that use additional summary statistics are deferred to Appendix Section F

**Compression into a window token.** We compress each presence vector into a single discrete *window token* using a pre-fitted Bernoulli mixture model with $K$ components. Let $z \in \{1,\dots,K\}$ denote the mixture component assignment for a window; we treat $z$ as the coarse observation token for that window. The mixture additionally yields a per-component Bernoulli parameter vector $\phi_z \in [0,1]^{|\mathcal{V}|}$, which can be interpreted as a probabilistic decoder over within-window vocab presence.

**Coarse autoregressive modeling.** The coarse trajectory is represented as a sequence over a discrete window-level vocabulary. Non-empty windows are assigned mixture-component tokens $z \in \{1,\dots,K\}$, while runs of empty windows are represented using gap tokens $\{\text{empty}_1,\dots,\text{empty}_{R_{\max}}\}$, where $\text{empty}_R$ denotes a gap of $R$ consecutive length-$\Delta$ windows with no observed events. Thus, the coarse autoregressive vocabulary is

$$\mathcal{V}_\Delta = \{1,\dots,K\} \cup \{\text{empty}_1,\dots,\text{empty}_{R_{\max}}\} \cup \mathcal{V}_{\text{special}},$$

where $\mathcal{V}_{\text{special}}$ includes the special sequence tokens EOS and a discharge anchor token.

Let $u_{j,1:T_\Delta}$ denote the resulting coarse token sequence, where each $u_{j,t}$ is either a non-empty window token $z$, an empty-window gap token, or a special token. We train a decoder-only autoregressive model

$$p_\theta(u_{j,1:T_\Delta}) = \prod_{t=1}^{T_\Delta} p_\theta(u_{j,t} \mid u_{j,<t}).$$

At inference, sampled empty-window tokens advance the generated timeline by the corresponding number of empty $\Delta$-windows. For each sampled non-empty token $z \in \{1,\dots,K\}$, we decode it into an EHR-like within-window realization by sampling a binary vocabulary-presence vector from the Bernoulli decoder $\phi_z \in [0,1]^{|\mathcal{V}|}$. Horizon risks are then computed by checking whether the task outcome $Y^{(k)}$ occurs in the decoded sampled trajectory within the corresponding horizon $H_k$.

### 4.3. Mixture of Resolutions (MoRGen): Task-Specific Mixture of Experts

Because different resolutions can be preferable in different regimes, we combine multiple single-resolution forecasters via a simple Mixture of Experts. Fix a set of resolutions $\{\Delta_1,\dots,\Delta_R\}$. For a given task and horizon, each resolution produces a probability $\hat{\pi}_j^{(r)} := \hat{\pi}_j(\Delta_r)$. We define a fused predictor

$$\hat{\pi}_j^{(\text{fuse})} := \sum_{r=1}^{R} w_r \, \hat{\pi}_j^{(r)}, \qquad w \in \Delta^{R-1},$$

where $w_r \geq 0$ and $\sum_{r=1}^{R} w_r = 1$.

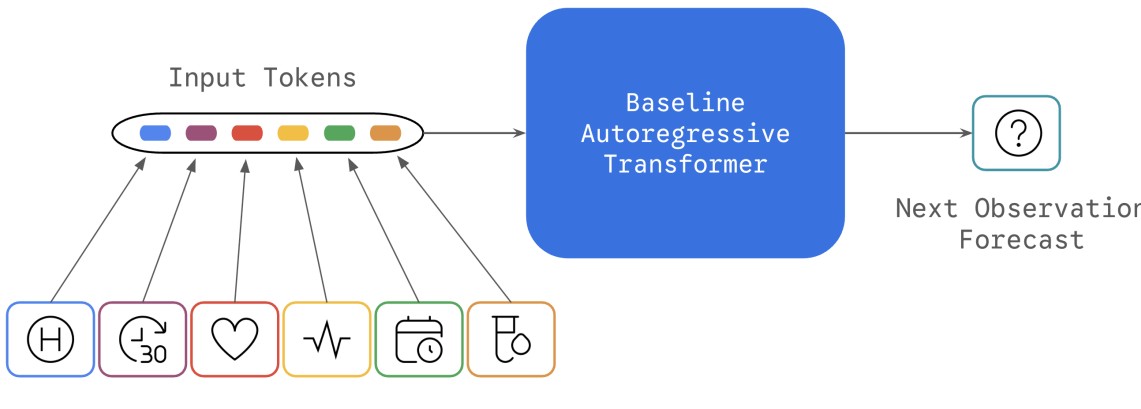

*Figure 1.* **Baseline single-resolution generative forecaster.** The model autoregressively generates time-gap tokens (minimum 1 minute) and observation tokens (codes and discretized numeric values), producing a sampled post-discharge trajectory. Horizon risks are computed from the fraction of sampled futures in which the outcome occurs within $H$.

**Learning mixture weights.** We learn $w$ on a validation split by minimizing BCE for the task/horizon:

$$\frac{1}{N_{\text{val}}} \sum_{j \in \text{val}} \Big[ -Y_j \log \hat{\pi}_j^{(\text{fuse})} - (1 - Y_j) \log(1 - \hat{\pi}_j^{(\text{fuse})}) \Big].$$

Importantly, $w$ is *shared across patients* (no per-patient gating), but is learned separately for each dataset, task, and horizon. This is a convex optimization task, and we use the SLSQP solver in SciPy (Virtanen et al., 2020) to efficiently solve this.

Because the mixture weights are fit using labeled validation outcomes, the main MoRGen predictor is not fully zero-shot end-to-end. Rather, MoRGen combines otherwise zero-shot generative expert forecasts with a low-capacity supervised calibration/fusion layer; we evaluate fully zero-shot uniform averaging and shared-weight variants in Appendix Section D.

## 5. Experiments: MoRGen Improves Performance, and Why

We run our experiments on three datasets: MIMIC-IV (Johnson et al., 2024), Hospital A (a private dataset of heart failure patients from a large tertiary care center), EHRSHOT (Wornow et al., 2023). On these datasets, we evaluate performance across multiple binary clinical forecasting tasks and horizons. The tasks evaluated across the three datasets are readmission, death, and abnormal laboratory value detection for leukocyte, platelet, hemoglobin, hematocrit, and creatinine. Specifically, creatinine corresponds to elevated creatinine > 2.3 mg/dL, while hematocrit, hemoglobin, leukocyte, and platelet correspond to decreased values < 24.6%,

< 8.0 g/dL, < 5.1 K/uL, and < 86.0 K/uL, respectively. Numeric-valued observations such as labs are discretized into deciles, whereas non-numeric events such as admissions are represented directly as discrete tokens. Accordingly, abnormal-value detection is defined on the discretized laboratory representation by checking whether the value associated with an observed laboratory decile ever exceeds or falls below the corresponding threshold by horizon H. See Appendix J for task, dataset, and additional implementation details.

We compare (i) a fine-grained baseline forecaster, (ii) a family of coarse-resolution forecasters at $\Delta \in$ {day, week, month, quarter, semi}, and (iii) MoRGen fusion over resolutions. For coarse models, performance depends on the window codebook size $K$, so we sweep four $K$ values (20 coarse models per dataset total). We report (a) **discrimination** via AUROC, (b) **calibration** via calibration error, and (c) **sharpness** via a proxy using average predictive confidence.

### 5.1. Consistent AUROC Gains from MoRGen

Tables 2 and 3 summarize AUROC for the baseline forecaster (fine-grained single resolution model). Across all datasets and both short (30-day) and long (2-year) horizons, MoRGen achieves statistically significant AUROC improvements[5]. Especially large gains are observed (a) across datasets for mortality and (b) in the smaller-data regime, as EHRSHOT has only 5,717 patients while MIMIC and Hospital A have 154,269 and 191,691 patients respectively.

---

[5]Tested via a one-tailed t-test with $\alpha = 0.05$.

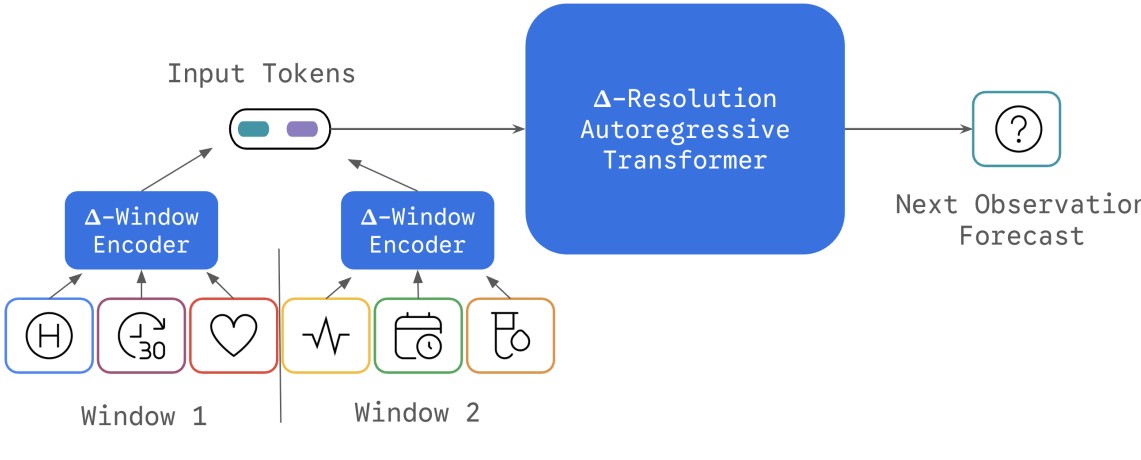

*Figure 2.* **Coarse-resolution forecasting via window tokenization.** Each length-$\Delta$ window is summarized by a vocabulary presence vector and compressed into a discrete window token via a Bernoulli mixture model. An autoregressive model is trained on the resulting window-token sequence.

*Table 2.* AUROC by outcome for 30-day horizon: Baseline vs. MoRGen across datasets. Entries report AUROC with jackknife standard errors over 50 sampled trajectories

| Outcome | MIMIC | | Hospital A | | EHRSHOT | |
|---|---|---|---|---|---|---|
| | Baseline | MoRGen | Baseline | MoRGen | Baseline | MoRGen |
| Creatinine | 92.16 ± 0.34 | **93.34 ± 0.21** | 94.50 ± 0.10 | **95.29 ± 0.05** | 77.85 ± 0.97 | **84.68 ± 0.58** |
| Death | 70.27 ± 0.85 | **89.06 ± 0.20** | 69.13 ± 0.44 | **81.67 ± 0.09** | 49.98 ± 0.02 | **81.46 ± 0.98** |
| Hematocrit | 88.21 ± 0.31 | **89.39 ± 0.22** | 91.29 ± 0.05 | **91.80 ± 0.03** | 79.08 ± 0.71 | **86.54 ± 0.30** |
| Hemoglobin | 88.15 ± 0.30 | **89.42 ± 0.23** | 91.63 ± 0.05 | **92.13 ± 0.03** | 76.93 ± 0.68 | **86.06 ± 0.28** |
| Leukocyte | 85.96 ± 0.17 | **86.31 ± 0.13** | 87.39 ± 0.05 | **87.73 ± 0.04** | 66.20 ± 0.81 | **81.06 ± 0.35** |
| Platelet | 93.91 ± 0.41 | **94.67 ± 0.20** | 93.49 ± 0.07 | **93.94 ± 0.05** | 83.60 ± 0.38 | **92.21 ± 0.16** |
| Readmission | 69.98 ± 0.20 | **70.28 ± 0.18** | 85.11 ± 0.06 | **85.50 ± 0.05** | 63.86 ± 0.55 | **72.35 ± 0.30** |

The rest of this section focuses on empirically explaining *why* multi-resolution fusion yields consistent gains: different resolutions exhibit task- and horizon-dependent tradeoffs in discrimination, calibration, and sharpness—all of which MoRGen exploits.

### 5.2. Optimal Temporal Resolution is Task and Horizon Dependent

A core empirical finding is that no single temporal resolution dominates across all tasks and horizons. Figure 4 illustrates three representative tasks where the optimal resolution (as measured by BCE) shifts substantially across datasets and horizons, ranging from week/day-level to quarter/semi-annual models. This motivates MoRGen as a *task and horizon specific* mixture: even when the minute-level baseline is strong on some tasks, there exist other tasks (and especially longer horizons) where coarser experts are more competitive.

### 5.3. Calibration Improves for Many Tasks Under Coarsening, While Sharpness Shifts Systematically

**Calibration and discrimination.** A consistent trend is that coarser resolutions often improve calibration error relative to the fine-grained baseline, even when discrimination differences are modest. Intuitively, temporal aggregation reduces sensitivity to low-frequency signals during generation and can regularize $\hat{\pi}$. To make this concrete, Table 4 reports the fraction of tasks for which the *best* coarse model (swept over $\Delta$ and $K$) improves calibration over the baseline at two representative horizons.

We observe more modest improvements in discrimination: the AUROC win rate ranges from $42.9\%$ at 30 days to $52.4\%$ at 2 years, suggesting that coarse experts are competitive on a substantial subset of task–horizon pairs. In contrast, calibration improvements hold for a large majority of tasks at both horizons. MoRGen's consistent AUROC gains, despite coarse experts only winning on a subset of task–horizon pairs, suggest that different resolutions learn complementary forecasting signals that the mixture can exploit.

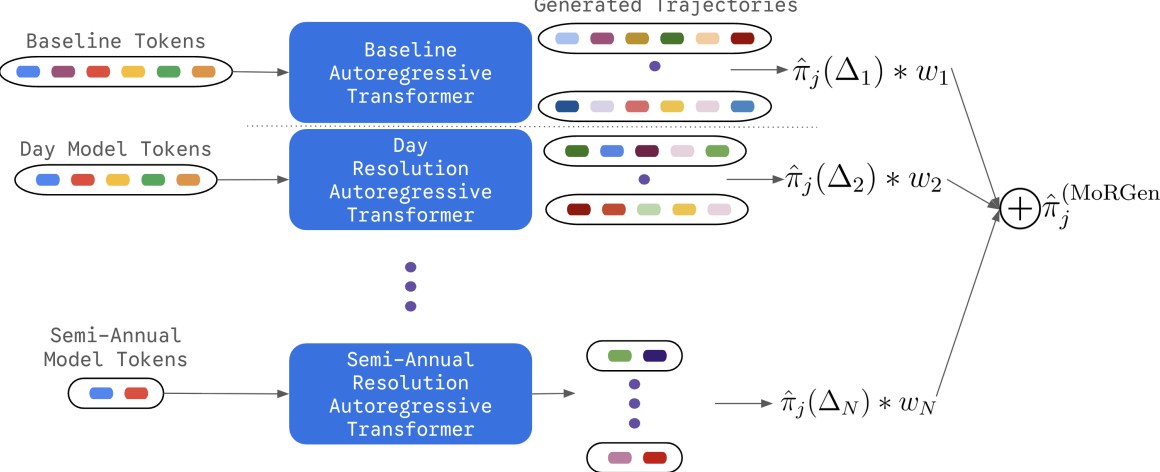

*Figure 3.* **MoRGen fusion.** Multiple resolution-specific forecasters produce horizon risks $\hat{\pi}_j(\Delta_r)$, which are combined via task/horizon-specific convex weights learned on validation data.

*Table 3.* AUROC by outcome for 2-year horizon: Baseline vs. MoRGen across datasets. Entries report AUROC with jackknife standard errors over 50 sampled trajectories

| | MIMIC | | Hospital A | | EHRSHOT | |
|---|---|---|---|---|---|---|
| Outcome | Baseline | MoRGen | Baseline | MoRGen | Baseline | MoRGen |
| Creatinine | 90.19 ± 0.15 | **90.77 ± 0.11** | 89.40 ± 0.08 | **90.41 ± 0.05** | 82.23 ± 0.49 | **86.30 ± 0.41** |
| Death | 59.22 ± 0.36 | **81.75 ± 0.06** | 58.93 ± 0.19 | **77.46 ± 0.04** | 49.98 ± 0.02 | **77.57 ± 0.26** |
| Hematocrit | 82.30 ± 0.12 | **83.45 ± 0.10** | 82.08 ± 0.05 | **83.85 ± 0.03** | 78.71 ± 0.38 | **83.71 ± 0.17** |
| Hemoglobin | 82.71 ± 0.11 | **84.04 ± 0.09** | 83.01 ± 0.05 | **84.66 ± 0.03** | 77.23 ± 0.38 | **82.86 ± 0.19** |
| Leukocyte | 83.03 ± 0.09 | **83.31 ± 0.06** | 80.60 ± 0.06 | **81.81 ± 0.03** | 62.71 ± 0.44 | **73.86 ± 0.27** |
| Platelet | 85.18 ± 0.27 | **85.82 ± 0.16** | 84.69 ± 0.06 | **85.45 ± 0.04** | 79.96 ± 0.33 | **86.53 ± 0.18** |
| Readmission | 68.32 ± 0.14 | **70.05 ± 0.09** | 73.38 ± 0.13 | **77.46 ± 0.05** | 65.72 ± 0.37 | **73.49 ± 0.10** |

*Table 4.* Win Rate for Discrimination and calibration error across different horizons $H$.

| Metric | H = 30 days (%) | H = 2 years (%) |
|---|---|---|
| AUROC | 42.9 | 52.4 |
| Calibration Error | 85.7 | 81.0 |

*Table 5.* Sharpness win rate across datasets. Each entry reports the percentage of tasks where a coarser model achieves higher sharpness than the minute-level baseline.

| Model | H = 30 days (%) | H = 2 years (%) |
|---|---|---|
| Day (1d) | 47.6 | 23.8 |
| Week (7d) | 76.2 | 61.9 |
| Month (30d) | 81.0 | 76.2 |
| Quarter (90d) | 85.7 | 66.7 |
| Semi (182d) | 90.5 | 66.7 |

**Sharpness (a confidence proxy).** In addition to calibration and discrimination, resolution affects how *confident* a forecaster tends to be. Higher sharpness corresponds to more extreme probabilities (predictions concentrated nearer 0 or 1), while lower sharpness corresponds to less decisive forecasts concentrated nearer 0.5. Importantly, sharpness is *not* a measure of correctness by itself: a model can be sharp and miscalibrated (overconfident), or less sharp but well-calibrated. We therefore interpret sharpness jointly with the win-rate trends in Table 4: coarser experts often improve calibration for many tasks, and Table 5 shows that they also frequently achieve higher sharpness. For example, the semi-annual model yields sharper predictions than the baseline for 90.5% of tasks at $H$=30 days and 66.7% at $H$=2 years; taken together with the calibration win rates,

this indicates coarse experts often provide complementary probability behavior that MoRGen can exploit via fusion.

**Takeaway.** Across tasks, changing $\Delta$ shifts calibration and sharpness in systematic, horizon-dependent ways. These shifts are *not* uniformly beneficial: coarsening can yield better-calibrated and more decisive probabilities for many task–horizon pairs, but can also degrade discrimination on many others. This motivates task/horizon-specific mixture weights: MoRGen can upweight resolutions that provide well-calibrated (and appropriately sharp) probabil-

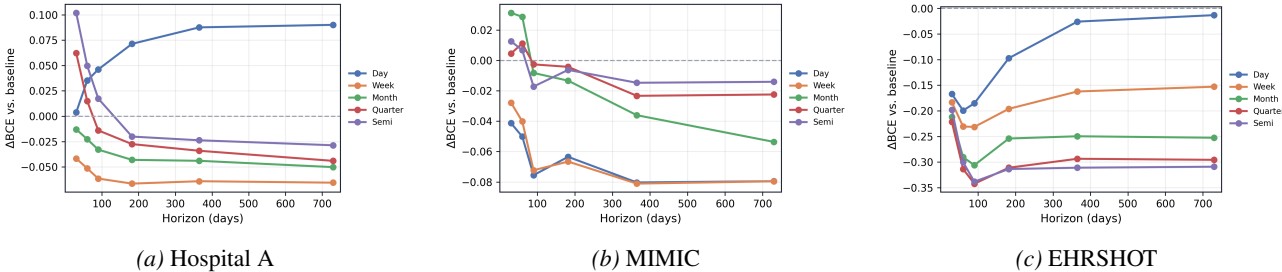

*Figure 4.* **Task-Specific Optimal Resolution Varies.** We plot horizon (in days) vs BCE and highlight examples where the best-performing resolution varies by task and dataset. Additional plots for all tasks and datasets are provided in Appendix L.

ities for a given task/horizon, while relying on finer resolutions when they are needed to preserve discriminative ranking.

### 5.4. Outcome-Specific Challenges: Mortality

Terminal outcomes such as mortality behave differently from laboratory abnormalities and readmission. Mortality is a one-shot event that often occurs far from discharge, and may be weakly determined by discharge history alone. In this regime, we observe that the minute-level baseline can be simultaneously poorly calibrated and insufficiently discriminative, while some coarser experts provide substantially improved probability estimates. This outcome-specific behavior helps explain why MoRGen yields particularly large gains on mortality across datasets and horizons (Tables 2 and 3).

### 5.5. Comparison to ETHOS

We additionally compare MoRGen to ETHOS (Renc et al., 2024; 2025), an autoregressive EHR trajectory generator, to assess whether multi-resolution fusion remains useful beyond our own fine-grained baseline. ETHOS is a strong comparator, outperforming the minute-level baseline in 30 of 42 matched task–horizon settings. However, MoRGen outperforms ETHOS in 27 of 42 settings, and incorporating ETHOS as an additional expert in the MoRGen mixture improves robustness, outperforming ETHOS alone in 40 of 42 settings. These results suggest that MoRGen is not tied to a particular autoregressive backbone, but can serve as a late-fusion framework for combining complementary generative forecasters. Full ETHOS results are provided in Section C.

## 6. Discussion and Limitations

Our experiments focus on post-discharge, long-horizon forecasting: we condition on inpatient history up to discharge and predict outcomes after discharge. Our results and analysis are limited to this regime, and it remains an open question whether (and at what level) temporal coarsening is benefi-

cial in the inpatient setting, where clinical state can evolve rapidly and fine-grained temporal structure may be more critical.

We do not explicitly model right-censoring at the evaluation horizons, and instead evaluate forecasts by whether they recapture observed future outcomes within each horizon. This choice simplifies analysis and matches common practice in recent zero-shot EHR generative forecasting work (Renc et al., 2024; 2025). Incorporating censoring-aware objectives and evaluation (e.g., survival-style likelihoods) is an important direction for future work, particularly for longer horizons where incomplete follow-up is very prevalent.

MoRGen learns a fixed population-level mixture for each dataset, task, and horizon, rather than adapting mixture weights to individual patients. Learning per-patient mixture weights is a natural future direction, since different patients may require different temporal resolutions depending on their history and queried endpoint.

Our coarse experts also use binary within-window presence vectors, which discard repeated event counts and event ordering within windows; this information loss may limit the performance of individual coarse autoregressive models.

MoRGen uses a late-fusion approach with separately trained autoregressive models; alternative hierarchical or multi-timescale architectures may be more performant and are promising directions for future work.

More broadly, methods aimed at improving long-sequence generation stability, such as diffusion-forcing-style approaches (Chen et al., 2024), are promising directions for improving long-range EHR forecasting.

## 7. Conclusion

We introduced MoRGen, a *mixture-of-resolutions* approach for clinical risk forecasting that fuses zero-shot generative expert forecasts across temporal resolutions. Our key empirical finding is that *temporal resolution is not one-size-fits-all*: the best operating point depends strongly on the outcome and prediction horizon, and coarse models often

exhibit substantially improved calibration behavior even when discrimination gains are modest. Motivated by this heterogeneity, MoRGen fuses multiple resolution-specific forecasters via a low-capacity mixture of experts fitted per task and horizon.

Across three independent datasets (MIMIC, Hospital A, and EHRSHOT), multiple outcomes, and both short (30-day) and long (2-year) horizons, MoRGen consistently improves AUROC over a strong fine-grained resolution baseline, with especially large gains (a) for mortality and (b) in the smaller-data regime. Through empirical experiments, we observe MoRGen tends to reduce calibration error, increase sharpness (confidence), and–on some endpoints–improve discrimination. These results provide an intuitive explanation for why multi-resolution fusion achieves consistent performance gains: i.e. different resolutions incur different and complementary errors that can be mitigated via ensembling.

More broadly, our results suggest that temporal coarsening can serve as an effective regularizer for generative forecasting, trading resolution for stability in a task- and horizon-dependent way, and that mixtures over resolutions provide a simple, scalable path to more robust forecasting performance.

## Impact Statement

Our experiments are retrospective and evaluate predictive performance metrics (AUROC, BCE, calibration error, etc.) rather than downstream clinical outcomes; thus, the results should not be interpreted as evidence of clinical benefit without prospective validation. We do not propose autonomous decision-making. Any real-world use would require rigorous validation across institutions and clinician interpretation of model predictions.

Future societal implications depend on how such models are deployed. Beneficial impacts include improved reliability of probabilistic forecasts enabling more accurate risk stratification of patients and earlier intervention. Potential harms include but are not limited to inequitable performance across patient groups and inaccuracy due to dataset and patient population shift. We encourage follow-up work on fairness auditing of generative clinical forecasting models and more rigorous assessments of the reliability of these forecasts in clinical contexts.

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

# A. Joint-Task Forecasting

**Appendix table abbreviations.**    To conserve table space, appendix tables abbreviate task names as follows: Creat. = abnormal creatinine, Hct = hematocrit, Hgb = abnormal hemoglobin, WBC = abnormal leukocyte / white blood cell count, Plt = abnormal platelet, Readm. = readmission. As stated in the main text, these abnormal lab tasks are defined as follows: creatinine corresponds to elevated creatinine $> 2.3$ mg/dL, while hematocrit, hemoglobin, leukocyte, and platelet correspond to decreased values $< 24.6\%$, $< 8.0$ g/dL, $< 5.1$ K/uL, and $< 86.0$ K/uL, respectively.

We evaluate whether multi-resolution forecasting remains beneficial for conjunctive endpoints, where the target is positive only if two clinical events both occur within their respective horizons. For each dataset, we form five endpoint pairs, fix the first event horizon at 30 days, and sweep the second event horizon over 90 days, 180 days, and 2 years. The evaluated pairs are creatinine/death, creatinine/platelet, hematocrit/readmission, leukocyte/death, and leukocyte/hemoglobin.

As shown in Table 6, MoRGen improves held-out AUROC over the minute-level baseline on every evaluated conjunctive endpoint across all three datasets. This corresponds to a 100% AUROC win rate across the 45 joint-task experiments. These results support that the gains from multi-resolution fusion are not limited to marginal single-event endpoint definitions; they also persist when endpoints require the joint occurrence of two events over different horizons.

*Table 6.* **Exact held-out joint-task AUROC results.** Each entry reports held-out AUROC; higher is better. The first event horizon is fixed at 30 days in every row. Within each horizon block, `Min` is the minute-level baseline and `MoR` is the full MoRGen fusion.

**MIMIC**

| Endpoint pair | 30d / 90d | | 30d / 180d | | 30d / 2y | |
| --- | --- | --- | --- | --- | --- | --- |
| | Min | MoR | Min | MoR | Min | MoR |
| Creat. & Death | $61.02 \pm 1.53$ | $87.98 \pm 0.53$ | $61.10 \pm 1.54$ | $88.63 \pm 0.43$ | $65.17 \pm 1.47$ | $92.20 \pm 0.21$ |
| Creat. & Plt | $84.22 \pm 1.78$ | $92.55 \pm 0.52$ | $85.76 \pm 1.57$ | $92.78 \pm 0.77$ | $88.40 \pm 1.65$ | $93.15 \pm 0.52$ |
| Hct & Readm. | $87.46 \pm 0.35$ | $88.57 \pm 0.26$ | $87.80 \pm 0.34$ | $88.92 \pm 0.28$ | $88.00 \pm 0.33$ | $89.13 \pm 0.27$ |
| WBC & Death | $59.60 \pm 1.52$ | $88.25 \pm 0.71$ | $62.52 \pm 1.25$ | $89.02 \pm 0.32$ | $61.01 \pm 0.95$ | $87.38 \pm 0.21$ |
| WBC & Hgb | $90.64 \pm 0.41$ | $91.62 \pm 0.21$ | $90.67 \pm 0.44$ | $91.45 \pm 0.23$ | $90.30 \pm 0.29$ | $91.14 \pm 0.16$ |

**Hospital A**

| Endpoint pair | 30d / 90d | | 30d / 180d | | 30d / 2y | |
| --- | --- | --- | --- | --- | --- | --- |
| | Min | MoR | Min | MoR | Min | MoR |
| Creat. & Death | $78.42 \pm 0.64$ | $90.14 \pm 0.07$ | $79.52 \pm 0.55$ | $90.49 \pm 0.06$ | $82.85 \pm 0.41$ | $90.83 \pm 0.06$ |
| Creat. & Plt | $89.25 \pm 0.28$ | $92.17 \pm 0.10$ | $89.44 \pm 0.27$ | $92.01 \pm 0.10$ | $90.08 \pm 0.22$ | $92.04 \pm 0.08$ |
| Hct & Readm. | $90.48 \pm 0.06$ | $91.07 \pm 0.04$ | $90.63 \pm 0.06$ | $91.21 \pm 0.04$ | $90.75 \pm 0.06$ | $91.30 \pm 0.03$ |
| WBC & Death | $73.51 \pm 0.62$ | $88.65 \pm 0.07$ | $73.82 \pm 0.50$ | $88.07 \pm 0.06$ | $74.44 \pm 0.34$ | $87.11 \pm 0.04$ |
| WBC & Hgb | $87.94 \pm 0.11$ | $88.81 \pm 0.06$ | $87.98 \pm 0.10$ | $88.81 \pm 0.05$ | $87.37 \pm 0.08$ | $87.90 \pm 0.04$ |

**EHRSHOT**

| Endpoint pair | 30d / 90d | | 30d / 180d | | 30d / 2y | |
| --- | --- | --- | --- | --- | --- | --- |
| | Min | MoR | Min | MoR | Min | MoR |
| Creat. & Death | $50.00 \pm 0.00$ | $82.70 \pm 2.00$ | $50.00 \pm 0.00$ | $84.38 \pm 1.39$ | $50.00 \pm 0.00$ | $74.25 \pm 1.59$ |
| Creat. & Plt | $80.93 \pm 2.27$ | $83.92 \pm 0.71$ | $81.88 \pm 2.16$ | $84.44 \pm 0.71$ | $82.93 \pm 1.93$ | $84.50 \pm 0.53$ |
| Hct & Readm. | $73.08 \pm 0.81$ | $84.91 \pm 0.30$ | $73.23 \pm 0.80$ | $85.41 \pm 0.27$ | $73.65 \pm 0.77$ | $85.70 \pm 0.27$ |
| WBC & Death | $49.98 \pm 0.02$ | $74.06 \pm 2.80$ | $49.98 \pm 0.02$ | $81.68 \pm 1.45$ | $49.98 \pm 0.02$ | $79.03 \pm 0.98$ |
| WBC & Hgb | $77.07 \pm 1.58$ | $86.92 \pm 1.06$ | $73.22 \pm 1.38$ | $85.66 \pm 1.05$ | $68.45 \pm 1.24$ | $83.18 \pm 0.84$ |

## B. Longer-Horizon Forecasting

We next evaluate whether the observed gains persist beyond the main paper's 2-year horizon. The extended horizons include 5-, 10-, and 20-year evaluations.

As shown in Table 7, MoRGen improves held-out AUROC over the minute-level baseline for every evaluated dataset, task, and added longer horizon. Across 3 datasets, 7 tasks, and 3 added horizons, this corresponds to a 100% AUROC win rate across 63 longer-horizon experiments. This suggests that the benefits of multi-resolution forecasting persist well beyond the 30-day and 2-year horizons emphasized in the main text.

*Table 7.* **Exact held-out AUROC results for the added longer-horizon evaluations.** Each entry reports held-out AUROC. Within each horizon block, `Min` is the minute-level baseline and `MoR` is the full MoRGen fusion.

**MIMIC**

| Task | 5y | | 10y | | 20y | |
| --- | --- | --- | --- | --- | --- | --- |
| | Min | MoR | Min | MoR | Min | MoR |
| Creat. | $87.80 \pm 0.14$ | $88.29 \pm 0.10$ | $86.53 \pm 0.14$ | $87.00 \pm 0.11$ | $86.40 \pm 0.15$ | $86.87 \pm 0.11$ |
| Death | $56.71 \pm 0.29$ | $80.24 \pm 0.07$ | $55.59 \pm 0.27$ | $78.24 \pm 0.08$ | $55.36 \pm 0.26$ | $77.64 \pm 0.08$ |
| Hct | $79.41 \pm 0.11$ | $80.83 \pm 0.08$ | $77.92 \pm 0.10$ | $79.21 \pm 0.09$ | $77.61 \pm 0.10$ | $78.89 \pm 0.08$ |
| Hgb | $79.49 \pm 0.12$ | $81.10 \pm 0.09$ | $77.44 \pm 0.11$ | $79.02 \pm 0.09$ | $77.12 \pm 0.10$ | $78.70 \pm 0.08$ |
| WBC | $81.27 \pm 0.10$ | $81.64 \pm 0.07$ | $79.94 \pm 0.09$ | $80.47 \pm 0.06$ | $79.80 \pm 0.09$ | $80.38 \pm 0.06$ |
| Plt | $81.09 \pm 0.26$ | $82.35 \pm 0.14$ | $79.89 \pm 0.25$ | $81.31 \pm 0.14$ | $79.57 \pm 0.24$ | $81.08 \pm 0.14$ |
| Readm. | $65.15 \pm 0.16$ | $68.12 \pm 0.09$ | $64.46 \pm 0.16$ | $68.41 \pm 0.09$ | $64.60 \pm 0.16$ | $68.73 \pm 0.09$ |

**Hospital A**

| Task | 5y | | 10y | | 20y | |
| --- | --- | --- | --- | --- | --- | --- |
| | Min | MoR | Min | MoR | Min | MoR |
| Creat. | $86.39 \pm 0.08$ | $87.40 \pm 0.05$ | $83.45 \pm 0.09$ | $84.36 \pm 0.05$ | $81.76 \pm 0.07$ | $82.69 \pm 0.04$ |
| Death | $53.38 \pm 0.14$ | $77.18 \pm 0.03$ | $52.22 \pm 0.15$ | $76.20 \pm 0.05$ | $54.05 \pm 0.14$ | $75.59 \pm 0.05$ |
| Hct | $77.67 \pm 0.05$ | $79.84 \pm 0.03$ | $74.06 \pm 0.05$ | $76.49 \pm 0.04$ | $71.92 \pm 0.06$ | $74.33 \pm 0.04$ |
| Hgb | $78.39 \pm 0.05$ | $80.60 \pm 0.03$ | $74.37 \pm 0.06$ | $76.81 \pm 0.04$ | $71.76 \pm 0.06$ | $74.18 \pm 0.04$ |
| WBC | $77.65 \pm 0.06$ | $79.19 \pm 0.03$ | $75.75 \pm 0.06$ | $77.57 \pm 0.03$ | $74.87 \pm 0.07$ | $76.85 \pm 0.03$ |
| Plt | $80.69 \pm 0.07$ | $81.66 \pm 0.04$ | $77.66 \pm 0.07$ | $78.77 \pm 0.04$ | $76.10 \pm 0.07$ | $77.12 \pm 0.04$ |
| Readm. | $66.47 \pm 0.22$ | $75.47 \pm 0.06$ | $62.07 \pm 0.24$ | $74.87 \pm 0.07$ | $60.54 \pm 0.22$ | $74.92 \pm 0.08$ |

**EHRSHOT**

| Task | 5y | | 10y | | 20y | |
| --- | --- | --- | --- | --- | --- | --- |
| | Min | MoR | Min | MoR | Min | MoR |
| Creat. | $82.29 \pm 0.45$ | $85.31 \pm 0.35$ | $82.63 \pm 0.42$ | $84.77 \pm 0.30$ | $83.66 \pm 0.38$ | $84.97 \pm 0.28$ |
| Death | $49.97 \pm 0.03$ | $73.96 \pm 0.14$ | $49.97 \pm 0.03$ | $73.90 \pm 0.18$ | $49.97 \pm 0.03$ | $73.75 \pm 0.18$ |
| Hct | $79.26 \pm 0.40$ | $82.55 \pm 0.18$ | $80.52 \pm 0.32$ | $82.83 \pm 0.16$ | $81.35 \pm 0.26$ | $83.08 \pm 0.16$ |
| Hgb | $77.85 \pm 0.39$ | $81.59 \pm 0.19$ | $78.90 \pm 0.36$ | $81.96 \pm 0.17$ | $79.62 \pm 0.32$ | $82.05 \pm 0.18$ |
| WBC | $63.46 \pm 0.41$ | $73.79 \pm 0.28$ | $66.50 \pm 0.40$ | $73.57 \pm 0.23$ | $67.32 \pm 0.41$ | $73.37 \pm 0.21$ |
| Plt | $80.53 \pm 0.33$ | $85.83 \pm 0.18$ | $81.88 \pm 0.30$ | $85.92 \pm 0.19$ | $83.05 \pm 0.30$ | $86.07 \pm 0.20$ |
| Readm. | $65.41 \pm 0.40$ | $70.96 \pm 0.12$ | $66.89 \pm 0.39$ | $72.28 \pm 0.14$ | $67.96 \pm 0.41$ | $72.36 \pm 0.13$ |

## C. Comparison to ETHOS

We additionally compare MoRGen to ETHOS (Renc et al., 2024; 2025), a peer-reviewed autoregressive EHR generation model that supports zero-shot risk estimation from sampled future trajectories. This comparison evaluates whether MoRGen's gains are specific to our fine-grained autoregressive baseline or whether multi-resolution fusion remains useful when compared against, and combined with, an external autoregressive generative model. For a controlled comparison, ETHOS is trained and evaluated on the same discretized observation vocabulary and downstream forecasting tasks used in our main experiments.

*Table 8.* **ETHOS comparison win-rate summary.** Each cell reports the number and percentage of task–horizon settings in which the first method achieves higher held-out AUROC than the second. The ETHOS+MoRGen comparison adds ETHOS as an additional expert for MoRGen to learn a weight for.

| Dataset | ETHOS > baseline | MoRGen > ETHOS | ETHOS+MoRGen > ETHOS |
|---|---|---|---|
| MIMIC | 9 / 14 (64.3%) | 8 / 14 (57.1%) | 12 / 14 (85.7%) |
| Hospital A | 14 / 14 (100.0%) | 8 / 14 (57.1%) | 14 / 14 (100.0%) |
| EHRSHOT | 7 / 14 (50.0%) | 11 / 14 (78.6%) | 14 / 14 (100.0%) |
| Overall | 30 / 42 (71.4%) | 27 / 42 (64.3%) | 40 / 42 (95.2%) |

As shown in Tables 8 and 9, ETHOS is a strong comparator and often improves over the minute-level baseline. MoRGen nevertheless outperforms ETHOS in a majority of the matched task–horizon settings. Moreover, adding ETHOS as an additional expert further improves robustness: ETHOS-augmented MoRGen outperforms ETHOS in 40 of 42 comparisons. This supports the interpretation of MoRGen as a general mixture framework that can incorporate complementary autoregressive experts, rather than as a method tied to one particular baseline architecture.

*Table 9.* **Exact held-out AUROC results for ETHOS comparisons.** Each entry reports held-out AUROC. Within each horizon block, `Base` is the minute-level baseline, `ETH` is ETHOS, `MoR` is the original MoRGen fusion, and `E+M` is the ETHOS-augmented MoRGen fusion described above.

| | | **MIMIC** | | | | | | |
|---|---|---|---|---|---|---|---|---|
| Task | | 30d | | | | 2y | | |
| | Base | ETH | MoR | E+M | Base | ETH | MoR | E+M |
| Creat. | $92.16 \pm 0.34$ | $91.18 \pm 0.44$ | $93.34 \pm 0.21$ | $93.63 \pm 0.18$ | $90.19 \pm 0.15$ | $89.58 \pm 0.25$ | $90.77 \pm 0.11$ | $91.35 \pm 0.09$ |
| Death | $70.27 \pm 0.85$ | $83.08 \pm 1.63$ | $89.06 \pm 0.20$ | $89.60 \pm 0.28$ | $59.22 \pm 0.36$ | $78.02 \pm 0.44$ | $81.75 \pm 0.06$ | $82.70 \pm 0.11$ |
| Hct | $88.21 \pm 0.31$ | $87.73 \pm 0.43$ | $89.39 \pm 0.22$ | $89.84 \pm 0.21$ | $82.30 \pm 0.12$ | $84.37 \pm 0.18$ | $83.45 \pm 0.10$ | $84.54 \pm 0.11$ |
| Hgb | $88.15 \pm 0.30$ | $87.06 \pm 0.58$ | $89.42 \pm 0.23$ | $89.73 \pm 0.21$ | $82.71 \pm 0.11$ | $84.22 \pm 0.21$ | $84.04 \pm 0.09$ | $84.70 \pm 0.12$ |
| WBC | $85.96 \pm 0.17$ | $87.20 \pm 0.29$ | $86.31 \pm 0.13$ | $87.42 \pm 0.19$ | $83.03 \pm 0.09$ | $85.29 \pm 0.13$ | $83.31 \pm 0.06$ | $85.25 \pm 0.11$ |
| Plt | $93.91 \pm 0.41$ | $93.63 \pm 0.38$ | $94.67 \pm 0.20$ | $95.31 \pm 0.15$ | $85.18 \pm 0.27$ | $85.42 \pm 0.57$ | $85.82 \pm 0.16$ | $86.98 \pm 0.18$ |
| Readm. | $69.98 \pm 0.20$ | $71.58 \pm 0.25$ | $70.28 \pm 0.18$ | $71.66 \pm 0.16$ | $68.32 \pm 0.14$ | $73.12 \pm 0.21$ | $70.05 \pm 0.09$ | $72.92 \pm 0.15$ |

| | | **Hospital A** | | | | | | |
|---|---|---|---|---|---|---|---|---|
| Task | | 30d | | | | 2y | | |
| | Base | ETH | MoR | E+M | Base | ETH | MoR | E+M |
| Creat. | $94.50 \pm 0.10$ | $95.01 \pm 0.13$ | $95.29 \pm 0.05$ | $95.79 \pm 0.05$ | $89.40 \pm 0.08$ | $90.37 \pm 0.11$ | $90.41 \pm 0.05$ | $91.09 \pm 0.06$ |
| Death | $69.13 \pm 0.44$ | $80.70 \pm 0.47$ | $81.67 \pm 0.09$ | $83.44 \pm 0.20$ | $58.93 \pm 0.19$ | $74.19 \pm 0.35$ | $77.46 \pm 0.04$ | $77.53 \pm 0.04$ |
| Hct | $91.29 \pm 0.05$ | $91.87 \pm 0.06$ | $91.80 \pm 0.03$ | $92.25 \pm 0.03$ | $82.08 \pm 0.05$ | $83.71 \pm 0.06$ | $83.85 \pm 0.03$ | $84.26 \pm 0.03$ |
| Hgb | $91.63 \pm 0.05$ | $92.05 \pm 0.06$ | $92.13 \pm 0.03$ | $92.52 \pm 0.03$ | $83.01 \pm 0.05$ | $84.45 \pm 0.07$ | $84.66 \pm 0.03$ | $85.04 \pm 0.03$ |
| WBC | $87.39 \pm 0.05$ | $88.16 \pm 0.10$ | $87.73 \pm 0.04$ | $88.70 \pm 0.05$ | $80.60 \pm 0.06$ | $82.55 \pm 0.14$ | $81.81 \pm 0.03$ | $83.45 \pm 0.05$ |
| Plt | $93.49 \pm 0.07$ | $93.74 \pm 0.12$ | $93.94 \pm 0.05$ | $94.62 \pm 0.05$ | $84.69 \pm 0.06$ | $86.17 \pm 0.12$ | $85.45 \pm 0.04$ | $86.91 \pm 0.07$ |
| Readm. | $85.11 \pm 0.06$ | $86.53 \pm 0.06$ | $85.50 \pm 0.05$ | $86.67 \pm 0.04$ | $73.38 \pm 0.13$ | $78.28 \pm 0.10$ | $77.46 \pm 0.05$ | $78.87 \pm 0.07$ |

| | | **EHRSHOT** | | | | | | |
|---|---|---|---|---|---|---|---|---|
| Task | | 30d | | | | 2y | | |
| | Base | ETH | MoR | E+M | Base | ETH | MoR | E+M |
| Creat. | $77.85 \pm 0.97$ | $81.96 \pm 2.06$ | $84.68 \pm 0.58$ | $87.35 \pm 0.59$ | $82.23 \pm 0.49$ | $81.57 \pm 1.51$ | $86.30 \pm 0.41$ | $87.75 \pm 0.30$ |
| Death | $49.98 \pm 0.02$ | $82.90 \pm 9.23$ | $81.46 \pm 0.98$ | $89.73 \pm 1.97$ | $49.98 \pm 0.02$ | $56.99 \pm 2.22$ | $77.57 \pm 0.26$ | $76.46 \pm 0.25$ |
| Hct | $79.08 \pm 0.71$ | $68.86 \pm 1.87$ | $86.54 \pm 0.30$ | $86.36 \pm 0.26$ | $78.71 \pm 0.38$ | $71.38 \pm 0.96$ | $83.71 \pm 0.17$ | $82.84 \pm 0.15$ |
| Hgb | $76.93 \pm 0.68$ | $67.57 \pm 2.22$ | $86.06 \pm 0.28$ | $86.01 \pm 0.26$ | $77.23 \pm 0.38$ | $70.73 \pm 1.19$ | $82.86 \pm 0.19$ | $81.88 \pm 0.19$ |
| WBC | $66.20 \pm 0.81$ | $91.56 \pm 1.42$ | $81.06 \pm 0.35$ | $92.60 \pm 1.43$ | $62.71 \pm 0.44$ | $82.93 \pm 0.80$ | $73.86 \pm 0.27$ | $85.99 \pm 0.57$ |
| Plt | $83.60 \pm 0.38$ | $89.40 \pm 1.73$ | $92.21 \pm 0.16$ | $93.72 \pm 0.16$ | $79.96 \pm 0.33$ | $82.83 \pm 2.30$ | $86.53 \pm 0.18$ | $87.16 \pm 0.17$ |
| Readm. | $63.86 \pm 0.55$ | $57.79 \pm 1.71$ | $72.35 \pm 0.30$ | $71.43 \pm 0.34$ | $65.72 \pm 0.37$ | $61.29 \pm 1.11$ | $73.49 \pm 0.10$ | $70.58 \pm 0.17$ |

## D. Truly Zero-Shot and Shared-Weight Fusion Ablations

The main MoRGen experiments learn convex mixture weights on labeled validation data separately for each dataset, task, and horizon. This provides a low-capacity supervised fusion layer over otherwise zero-shot generative experts, but it is not itself fully zero-shot. We therefore evaluate two restricted fusion variants that test whether the gains from multi-resolution experts persist without highly task-specific mixture fitting.

First, we evaluate a truly zero-shot MoRGen variant that uniformly averages the expert predictions, without using labeled validation data to learn mixture weights. This is a deliberately simple test of whether the multi-resolution expert pool is useful even before task-specific calibration. Second, we evaluate a shared-weight convex fusion variant, where a single set of mixture weights is learned jointly across all tasks and both headline horizons within each dataset. This still uses labeled validation data, but removes task- and horizon-specific fitting.

*Table 10.* **Fusion ablation win-rate summary at 30 days and 2 years.** Each row reports how often the corresponding fusion variant achieves higher held-out AUROC than the minute-level baseline across 3 datasets, 7 tasks, and 2 horizons.

| Variant | Wins / total | Win rate |
|---|---|---|
| Task/horizon-specific learned fusion | 42 / 42 | 100.0% |
| Uniform averaging | 28 / 42 | 66.7% |
| Shared-weight convex fusion | 42 / 42 | 100.0% |

As shown in Table 10, uniform averaging improves over the minute-level baseline in 28 of 42 experiments. This is a strong result for a simple truly zero-shot MoRGen variant: without any task-specific labels or fitted mixture weights, averaging across independently trained resolution experts is already better than the fine-grained baseline in two thirds of the evaluated settings. The shared-weight variant improves over the baseline in every evaluated setting, indicating that the gains are not solely due to fitting separate mixture weights for each task and horizon.

# E. Sensitivity to the Window Codebook Size

The full MoRGen model fuses across multiple temporal resolutions and multiple codebook sizes, allowing the validation objective to select among a broad pool of coarse experts. We evaluate a stricter single-$K$ ablation to test whether performance is sensitive to the exact codebook-size sweep. In this ablation, MoRGen is restricted to the minute-level baseline plus coarse experts from one codebook size at a time.

*Table 11.* **Single-$K$ MoRGen win-rate summary at 30 days and 2 years.** Each comparison tests whether a single-$K$ MoRGen variant achieves higher held-out AUROC than the minute-level baseline.

| Variant | Wins / total | Win rate |
|---|---|---|
| Single-$K$ MoRGen variants | 167 / 168 | 99.4% |

Across all datasets, tasks, headline horizons, and codebook sizes, the single-$K$ variants outperform the minute-level baseline in 167 of 168 experiments. This indicates that MoRGen's gains are not driven by a narrow or fragile choice of $K$. Although the full model retains the codebook-size sweep because it is inexpensive and provides additional robustness, the single-$K$ ablation suggests that the main benefit comes from combining temporal resolutions rather than from carefully tuning a particular codebook size.

*Table 12.* **EHRSHOT $K$-ablation AUROC at 30 days.** Each row reports the AUROC on the held-out set. The full MoRGen row uses all codebook sizes (all K), while each single-$K$ row leaves only one codebook size (one K) active in the fusion pool.

| Model | Creat. | Death | Hct | Hgb | WBC | Plt | Readm. |
|---|---|---|---|---|---|---|---|
| Baseline | 77.85 ± 0.97 | 49.98 ± 0.02 | 79.08 ± 0.71 | 76.93 ± 0.68 | 66.20 ± 0.81 | 83.60 ± 0.38 | 63.86 ± 0.55 |
| MoRGen | **84.68 ± 0.58** | 81.46 ± 0.98 | **86.54 ± 0.30** | **86.06 ± 0.28** | **81.06 ± 0.35** | **92.21 ± 0.16** | **72.35 ± 0.30** |
| MoRGen 64 | 81.27 ± 0.71 | 79.42 ± 1.54 | 86.35 ± 0.26 | 85.55 ± 0.35 | 79.02 ± 0.64 | 91.28 ± 0.21 | 71.17 ± 0.45 |
| MoRGen 128 | 82.20 ± 0.59 | 82.36 ± 0.89 | 85.51 ± 0.37 | 85.23 ± 0.38 | 76.06 ± 0.44 | 90.48 ± 0.24 | 70.79 ± 0.33 |
| MoRGen 256 | 82.20 ± 0.67 | 82.73 ± 1.17 | 86.11 ± 0.24 | 85.32 ± 0.26 | 80.06 ± 0.43 | 91.32 ± 0.14 | 68.68 ± 0.34 |
| MoRGen 512 | 84.47 ± 0.53 | **83.41 ± 1.08** | 86.03 ± 0.36 | 85.67 ± 0.35 | 72.61 ± 0.40 | 92.16 ± 0.18 | 69.76 ± 0.33 |

*Table 13.* **EHRSHOT $K$-ablation AUROC at 2 years.** Each row reports the AUROC on the held-out set. The full MoRGen row uses all codebook sizes (all K), while each single-$K$ row leaves only one codebook size (one K) active in the fusion pool.

| Model | Creat. | Death | Hct | Hgb | WBC | Plt | Readm. |
|---|---|---|---|---|---|---|---|
| Baseline | 82.23 ± 0.49 | 49.98 ± 0.02 | 78.71 ± 0.38 | 77.23 ± 0.38 | 62.71 ± 0.44 | 79.96 ± 0.33 | 65.72 ± 0.37 |
| MoRGen | **86.30 ± 0.41** | **77.57 ± 0.26** | **83.71 ± 0.17** | **82.86 ± 0.19** | 73.86 ± 0.27 | **86.53 ± 0.18** | **73.49 ± 0.10** |
| MoRGen 64 | 83.88 ± 0.47 | 71.81 ± 0.48 | 83.18 ± 0.22 | 82.43 ± 0.20 | 73.87 ± 0.48 | 83.53 ± 0.28 | 68.92 ± 0.22 |
| MoRGen 128 | 84.77 ± 0.41 | 74.99 ± 0.37 | 82.38 ± 0.18 | 81.33 ± 0.23 | 73.79 ± 0.38 | 85.28 ± 0.21 | 70.35 ± 0.19 |
| MoRGen 256 | 85.68 ± 0.44 | 74.56 ± 0.36 | 82.27 ± 0.24 | 81.57 ± 0.22 | **73.89 ± 0.38** | 84.74 ± 0.21 | 68.50 ± 0.19 |
| MoRGen 512 | 85.43 ± 0.37 | 75.68 ± 0.38 | 82.39 ± 0.20 | 81.85 ± 0.18 | 69.44 ± 0.30 | 86.14 ± 0.20 | 69.01 ± 0.19 |

*Table 14.* **MIMIC $K$-ablation AUROC at 30 days.** Each row reports the AUROC on the held-out set. The full MoRGen row uses all codebook sizes (all K), while each single-$K$ row leaves only one codebook size (one K) active in the fusion pool.

| Model | Creat. | Death | Hct | Hgb | WBC | Plt | Readm. |
|---|---|---|---|---|---|---|---|
| Baseline | 92.16 ± 0.34 | 70.27 ± 0.85 | 88.21 ± 0.31 | 88.15 ± 0.30 | 85.96 ± 0.17 | 93.91 ± 0.41 | 69.98 ± 0.20 |
| MoRGen | 93.34 ± 0.21 | **89.06 ± 0.20** | **89.39 ± 0.22** | **89.42 ± 0.23** | **86.31 ± 0.13** | 94.67 ± 0.20 | **70.28 ± 0.18** |
| MoRGen 2048 | **93.43 ± 0.17** | 88.15 ± 0.27 | 89.19 ± 0.23 | 89.30 ± 0.21 | 86.28 ± 0.15 | 94.71 ± 0.28 | 70.26 ± 0.18 |
| MoRGen 4096 | 93.30 ± 0.22 | 87.95 ± 0.30 | 89.17 ± 0.23 | 89.28 ± 0.22 | 86.26 ± 0.13 | **94.75 ± 0.31** | 70.22 ± 0.19 |
| MoRGen 8192 | 93.05 ± 0.24 | 88.38 ± 0.28 | 88.87 ± 0.26 | 88.99 ± 0.26 | 86.22 ± 0.14 | 94.58 ± 0.21 | 70.22 ± 0.18 |
| MoRGen 16384 | 93.05 ± 0.21 | 88.46 ± 0.26 | 89.08 ± 0.26 | 89.09 ± 0.27 | 86.18 ± 0.13 | 94.53 ± 0.24 | 70.27 ± 0.19 |

*Table 15.* **MIMIC $K$-ablation AUROC at 2 years.** Each row reports the AUROC on the held-out set. The full MoRGen row uses all codebook sizes (all K), while each single-$K$ row leaves only one codebook size (one K) active in the fusion pool.

| Model | Creat. | Death | Hct | Hgb | WBC | Plt | Readm. |
|---|---|---|---|---|---|---|---|
| Baseline | 90.19 ± 0.15 | 59.22 ± 0.36 | 82.30 ± 0.12 | 82.71 ± 0.11 | 83.03 ± 0.09 | 85.18 ± 0.27 | 68.32 ± 0.14 |
| MoRGen | **90.77 ± 0.11** | **81.75 ± 0.06** | 83.45 ± 0.10 | 84.04 ± 0.09 | 83.31 ± 0.06 | 85.82 ± 0.16 | **70.05 ± 0.09** |
| MoRGen 2048 | 90.75 ± 0.11 | 81.10 ± 0.10 | **83.46 ± 0.10** | **84.06 ± 0.09** | 83.03 ± 0.07 | 85.85 ± 0.17 | 70.01 ± 0.10 |
| MoRGen 4096 | 90.75 ± 0.11 | 81.44 ± 0.07 | 83.29 ± 0.10 | 83.89 ± 0.10 | 83.23 ± 0.07 | 85.90 ± 0.17 | 69.87 ± 0.10 |
| MoRGen 8192 | 90.65 ± 0.11 | 81.39 ± 0.08 | 82.99 ± 0.10 | 83.56 ± 0.09 | 83.17 ± 0.07 | 85.64 ± 0.17 | 69.92 ± 0.10 |
| MoRGen 16384 | 90.72 ± 0.11 | 81.30 ± 0.08 | 83.22 ± 0.11 | 83.78 ± 0.10 | **83.36 ± 0.07** | **85.98 ± 0.18** | 69.94 ± 0.10 |

## F. Count-Augmented Window Encodings

The main coarse-resolution representation summarizes each fixed-length window using a binary presence vector, indicating whether each vocabulary item appears at least once in the window. This representation is intentionally simple, but it discards repeated within-window occurrences. Repeated events over a short period can be clinically meaningful, so we evaluate whether count-aware window summaries improve forecasting.

We compare the original binary MoRGen representation to two count-capped variants on EHRSHOT. The count-capped variants replace binary within-window indicators with histogram-style summaries in which event counts are clipped at either 2 or 64 before quantization. Because these count-valued vectors are no longer binary, we use vector quantization (Van Den Oord et al., 2017) for these variants rather than the Bernoulli-mixture EM procedure used for the binary presence vectors. This provides a direct test of whether retaining within-window count information improves downstream MoRGen forecasts.

As shown in Table 18, the count-capped variants do not improve over the binary presence representation in this setting. The binary MoRGen representation achieves the highest AUROC for all seven EHRSHOT tasks at 2 years and for six of seven tasks at 30 days. The one exception is 30-day mortality, where Count-64 slightly exceeds the binary representation. Overall, these fused-model results suggest that the simple binary window-presence representation is a strong default for the evaluated post-discharge forecasting tasks.

To further inspect whether this pattern is driven by the fused model or by the underlying experts, we also report representative individual-expert results at $K = 256$ for each window encoding and temporal resolution. These tables are intended to summarize the per-expert behavior without reproducing the full sweep over all codebook sizes. The fused results above use the corresponding full expert pools for each representation, rather than only the $K = 256$ experts shown below.

The individual-expert results are consistent with the fused comparison. Across the 70 matched resolution–task–horizon AUROC comparisons at $K = 256$, the binary experts outperform the Count-2 experts in 64 / 70 cases (91.4%) and the Count-64 experts in 70 / 70 cases (100.0%). For BCE, where lower values are better, the binary experts outperform Count-2 in 43 / 70 cases (61.4%) and Count-64 in 50 / 70 cases (71.4%). Thus, count-capped encodings occasionally improve BCE for individual task–resolution pairs, but they do not consistently improve either AUROC or fused MoRGen performance. These results support the use of binary window-presence vectors and Bernoulli-mixture EM in the main model. For binary window summaries, the Bernoulli likelihood is well matched to the representation, while count-valued summaries require a different quantization procedure and did not improve performance in this comparison.

*Table 16.* **Hospital A $K$-ablation AUROC at 30 days.** Each row reports the AUROC on the held-out set. The full MoRGen row uses all codebook sizes (all K), while each single-$K$ row leaves only one codebook size (one K) active in the fusion pool.

| Model | Creat. | Death | Hct | Hgb | WBC | Plt | Readm. |
|---|---|---|---|---|---|---|---|
| Baseline | $94.50 \pm 0.10$ | $69.13 \pm 0.44$ | $91.29 \pm 0.05$ | $91.63 \pm 0.05$ | $87.39 \pm 0.05$ | $93.49 \pm 0.07$ | $85.11 \pm 0.06$ |
| MoRGen | $95.29 \pm 0.05$ | $\mathbf{81.67 \pm 0.09}$ | $\mathbf{91.80 \pm 0.03}$ | $\mathbf{92.13 \pm 0.03}$ | $87.73 \pm 0.04$ | $\mathbf{93.94 \pm 0.05}$ | $\mathbf{85.50 \pm 0.05}$ |
| MoRGen 2048 | $95.27 \pm 0.05$ | $80.71 \pm 0.13$ | $91.68 \pm 0.04$ | $92.01 \pm 0.04$ | $87.61 \pm 0.05$ | $93.84 \pm 0.05$ | $85.42 \pm 0.05$ |
| MoRGen 4096 | $95.28 \pm 0.05$ | $80.71 \pm 0.12$ | $91.70 \pm 0.04$ | $92.03 \pm 0.03$ | $87.67 \pm 0.04$ | $93.86 \pm 0.05$ | $85.46 \pm 0.05$ |
| MoRGen 8192 | $\mathbf{95.30 \pm 0.05}$ | $80.06 \pm 0.14$ | $91.66 \pm 0.04$ | $92.00 \pm 0.03$ | $87.69 \pm 0.04$ | $93.92 \pm 0.05$ | $85.40 \pm 0.06$ |
| MoRGen 16384 | $95.25 \pm 0.05$ | $80.45 \pm 0.14$ | $91.70 \pm 0.04$ | $92.05 \pm 0.04$ | $\mathbf{87.73 \pm 0.04}$ | $93.93 \pm 0.05$ | $85.40 \pm 0.05$ |

*Table 17.* **Hospital A $K$-ablation AUROC at 2 years.** Each row reports the AUROC on the held-out set. The full MoRGen row uses all codebook sizes (all K), while each single-$K$ row leaves only one codebook size (one K) active in the fusion pool.

| Model | Creat. | Death | Hct | Hgb | WBC | Plt | Readm. |
|---|---|---|---|---|---|---|---|
| Baseline | $89.40 \pm 0.08$ | $58.93 \pm 0.19$ | $82.08 \pm 0.05$ | $83.01 \pm 0.05$ | $80.60 \pm 0.06$ | $84.69 \pm 0.06$ | $73.38 \pm 0.13$ |
| MoRGen | $\mathbf{90.41 \pm 0.05}$ | $\mathbf{77.46 \pm 0.04}$ | $\mathbf{83.85 \pm 0.03}$ | $\mathbf{84.66 \pm 0.03}$ | $81.81 \pm 0.03$ | $\mathbf{85.45 \pm 0.04}$ | $\mathbf{77.46 \pm 0.05}$ |
| MoRGen 2048 | $90.13 \pm 0.06$ | $76.89 \pm 0.06$ | $83.64 \pm 0.04$ | $84.55 \pm 0.03$ | $80.95 \pm 0.04$ | $85.04 \pm 0.04$ | $76.99 \pm 0.07$ |
| MoRGen 4096 | $90.23 \pm 0.06$ | $76.85 \pm 0.05$ | $83.65 \pm 0.03$ | $84.51 \pm 0.03$ | $81.29 \pm 0.04$ | $85.26 \pm 0.04$ | $76.87 \pm 0.08$ |
| MoRGen 8192 | $90.38 \pm 0.06$ | $76.64 \pm 0.05$ | $83.61 \pm 0.04$ | $84.42 \pm 0.03$ | $81.45 \pm 0.04$ | $85.35 \pm 0.04$ | $76.88 \pm 0.06$ |
| MoRGen 16384 | $90.35 \pm 0.06$ | $76.96 \pm 0.05$ | $83.41 \pm 0.03$ | $84.31 \pm 0.03$ | $\mathbf{81.82 \pm 0.03}$ | $85.41 \pm 0.04$ | $76.86 \pm 0.08$ |

*Table 18.* **EHRSHOT AUROC comparison for count-capped window encodings.** Each entry reports held-out AUROC. The original binary MoRGen results use Bernoulli window-presence vectors. Count-2 and Count-64 use count-capped window summaries with counts clipped at 2 and 64, respectively, followed by vector quantization. All values report held-out AUROC for the full convex-fusion MoRGen evaluator.

| Task | 30d | | | 2y | | |
|---|---|---|---|---|---|---|
| | Binary MoR | Count-2 MoR | Count-64 MoR | Binary MoR | Count-2 MoR | Count-64 MoR |
| Creat. | $84.68 \pm 0.58$ | $77.52 \pm 0.92$ | $77.38 \pm 0.94$ | $86.30 \pm 0.41$ | $82.22 \pm 0.46$ | $82.31 \pm 0.52$ |
| Death | $81.46 \pm 0.98$ | $75.25 \pm 2.32$ | $82.38 \pm 1.68$ | $77.57 \pm 0.26$ | $73.73 \pm 0.37$ | $69.59 \pm 0.33$ |
| Hct | $86.54 \pm 0.30$ | $80.75 \pm 0.56$ | $79.52 \pm 0.63$ | $83.71 \pm 0.17$ | $80.34 \pm 0.33$ | $79.92 \pm 0.35$ |
| Hgb | $86.06 \pm 0.28$ | $79.35 \pm 0.52$ | $77.98 \pm 0.58$ | $82.86 \pm 0.19$ | $79.22 \pm 0.33$ | $78.09 \pm 0.35$ |
| WBC | $81.06 \pm 0.35$ | $72.79 \pm 0.52$ | $71.88 \pm 0.84$ | $73.86 \pm 0.27$ | $64.20 \pm 0.51$ | $62.41 \pm 0.45$ |
| Plt | $92.21 \pm 0.16$ | $87.21 \pm 0.28$ | $83.94 \pm 0.37$ | $86.53 \pm 0.18$ | $81.92 \pm 0.33$ | $80.25 \pm 0.35$ |
| Readm. | $72.35 \pm 0.30$ | $70.50 \pm 0.33$ | $71.24 \pm 0.37$ | $73.49 \pm 0.10$ | $72.09 \pm 0.21$ | $72.08 \pm 0.30$ |

*Table 19.* **EHRSHOT K=256 individual-expert AUROC comparison for binary and count-capped window encodings.** Each entry reports held-out AUROC; higher is better. `Bin` uses binary window-presence vectors, while `C2` and `C64` use count-capped window encodings with counts clipped at 2 and 64, respectively. Strategy suffixes `D/W/Mo/Q/S` denote day/week/month/quarter/semi-annual experts.

| Strategy | 30d | | | | | | | 730d | | | | | | |
|---|---|---|---|---|---|---|---|---|---|---|---|---|---|---|
| | Creat. | Death | Hct | Hgb | WBC | Plt | Readm. | Creat. | Death | Hct | Hgb | WBC | Plt | Readm. |
| Bin-D | 76.95 | 80.92 | 84.04 | 83.85 | 76.74 | 88.94 | 68.58 | 73.01 | 67.66 | 79.25 | 79.02 | 65.28 | 80.84 | 69.34 |
| C2-D | 67.31 | 62.22 | 77.28 | 76.63 | 61.50 | 81.42 | 68.41 | 61.88 | 50.77 | 63.03 | 63.30 | 67.33 | 58.38 | 64.61 |
| C64-D | 63.00 | 78.62 | 78.96 | 79.50 | 63.48 | 82.37 | 68.12 | 60.77 | 64.09 | 60.63 | 60.21 | 62.12 | 60.69 | 62.31 |
| Bin-W | 78.57 | 78.97 | 81.54 | 80.31 | 69.57 | 88.98 | 64.53 | 78.95 | 72.29 | 76.27 | 75.87 | 63.60 | 80.16 | 65.88 |
| C2-W | 65.88 | 67.00 | 73.05 | 74.81 | 66.56 | 78.37 | 63.43 | 66.63 | 66.61 | 65.88 | 66.03 | 66.61 | 64.36 | 64.70 |
| C64-W | 58.30 | 73.31 | 71.00 | 69.41 | 63.56 | 75.21 | 62.49 | 61.41 | 68.09 | 63.58 | 63.96 | 61.70 | 64.52 | 63.62 |
| Bin-Mo | 76.89 | 77.42 | 80.60 | 79.16 | 72.45 | 88.03 | 66.15 | 79.64 | 72.04 | 75.95 | 75.70 | 66.08 | 79.62 | 67.96 |
| C2-Mo | 64.79 | 60.58 | 69.43 | 63.31 | 66.88 | 73.79 | 62.42 | 65.12 | 69.37 | 64.94 | 62.56 | 66.17 | 64.03 | 64.51 |
| C64-Mo | 57.34 | 65.19 | 72.61 | 71.10 | 65.39 | 75.89 | 63.36 | 59.26 | 62.88 | 61.05 | 62.01 | 60.88 | 61.49 | 63.22 |
| Bin-Q | 76.77 | 72.74 | 77.75 | 78.60 | 79.13 | 84.63 | 61.92 | 76.92 | 71.56 | 74.20 | 73.43 | 70.77 | 77.25 | 66.49 |
| C2-Q | 63.20 | 71.57 | 71.19 | 72.72 | 64.13 | 75.82 | 63.27 | 67.10 | 69.70 | 65.21 | 66.44 | 63.28 | 67.49 | 66.63 |
| C64-Q | 64.03 | 71.13 | 69.68 | 71.92 | 62.57 | 66.80 | 60.52 | 62.07 | 64.84 | 60.89 | 62.47 | 60.01 | 58.95 | 60.94 |
| Bin-S | 80.55 | 74.99 | 75.61 | 74.75 | 73.15 | 80.36 | 60.46 | 73.19 | 72.27 | 72.79 | 73.58 | 67.43 | 75.00 | 64.84 |
| C2-S | 63.98 | 72.05 | 66.72 | 69.94 | 68.13 | 69.37 | 60.75 | 62.10 | 66.48 | 63.70 | 66.50 | 61.18 | 65.53 | 63.73 |
| C64-S | 59.03 | 74.33 | 65.00 | 64.05 | 60.09 | 64.60 | 60.01 | 56.82 | 62.91 | 56.59 | 59.12 | 57.03 | 57.04 | 60.88 |

*Table 20.* **EHRSHOT K=256 individual-expert BCE comparison for binary and count-capped window encodings.** Each entry reports held-out BCE; lower is better. `Bin` uses binary window-presence vectors, while `C2` and `C64` use count-capped window encodings with counts clipped at 2 and 64, respectively. Strategy suffixes `D/W/Mo/Q/S` denote day/week/month/quarter/semi-annual experts.

| Strategy | 30d | | | | | | | 730d | | | | | | |
|---|---|---|---|---|---|---|---|---|---|---|---|---|---|---|
| | Creat. | Death | Hct | Hgb | WBC | Plt | Readm. | Creat. | Death | Hct | Hgb | WBC | Plt | Readm. |
| Bin-D | 0.289 | 0.160 | 0.259 | 0.246 | 0.297 | 0.257 | 0.436 | 0.990 | 0.692 | 0.523 | 0.542 | 1.028 | 0.641 | 0.713 |
| C2-D | 0.194 | 0.092 | 0.266 | 0.239 | 0.353 | 0.346 | 0.432 | 0.913 | 0.455 | 0.819 | 0.795 | 1.306 | 1.299 | 0.816 |
| C64-D | 0.314 | 0.145 | 0.341 | 0.331 | 0.857 | 0.483 | 0.475 | 1.263 | 0.795 | 1.069 | 1.100 | 1.752 | 1.319 | 0.856 |
| Bin-W | 0.138 | 0.076 | 0.265 | 0.258 | 0.276 | 0.237 | 0.479 | 0.307 | 0.425 | 0.538 | 0.533 | 0.601 | 0.438 | 0.710 |
| C2-W | 0.142 | 0.097 | 0.283 | 0.241 | 0.270 | 0.342 | 0.472 | 0.472 | 0.436 | 0.571 | 0.553 | 0.871 | 0.711 | 0.677 |
| C64-W | 0.155 | 0.091 | 0.280 | 0.244 | 0.448 | 0.346 | 0.459 | 0.555 | 0.413 | 0.608 | 0.617 | 1.430 | 0.782 | 0.681 |
| Bin-Mo | 0.130 | 0.091 | 0.326 | 0.289 | 0.284 | 0.264 | 0.583 | 0.256 | 0.444 | 0.796 | 0.709 | 0.501 | 0.471 | 0.809 |
| C2-Mo | 0.140 | 0.103 | 0.313 | 0.279 | 0.260 | 0.392 | 0.513 | 0.368 | 0.394 | 0.535 | 0.525 | 0.764 | 0.563 | 0.679 |
| C64-Mo | 0.144 | 0.089 | 0.298 | 0.249 | 0.310 | 0.360 | 0.485 | 0.409 | 0.396 | 0.557 | 0.551 | 1.171 | 0.674 | 0.672 |
| Bin-Q | 0.125 | 0.083 | 0.285 | 0.239 | 0.211 | 0.311 | 0.514 | 0.257 | 0.404 | 0.555 | 0.539 | 0.465 | 0.449 | 0.751 |
| C2-Q | 0.146 | 0.091 | 0.341 | 0.264 | 0.249 | 0.401 | 0.488 | 0.318 | 0.400 | 0.549 | 0.510 | 0.679 | 0.530 | 0.671 |
| C64-Q | 0.147 | 0.089 | 0.350 | 0.289 | 0.258 | 0.447 | 0.571 | 0.329 | 0.392 | 0.553 | 0.526 | 0.890 | 0.555 | 0.697 |
| Bin-S | 0.134 | 0.091 | 0.323 | 0.263 | 0.256 | 0.409 | 0.598 | 0.268 | 0.427 | 0.566 | 0.515 | 0.444 | 0.479 | 0.776 |
| C2-S | 0.153 | 0.102 | 0.363 | 0.287 | 0.249 | 0.447 | 0.548 | 0.312 | 0.472 | 0.565 | 0.518 | 0.582 | 0.526 | 0.693 |
| C64-S | 0.162 | 0.092 | 0.390 | 0.323 | 0.253 | 0.503 | 0.613 | 0.317 | 0.404 | 0.596 | 0.562 | 0.713 | 0.555 | 0.731 |

## G. Runtime Measurements

We benchmark the computational cost of MoRGen relative to the minute-level baseline. Because MoRGen combines multiple independently trained experts, inference can be performed either sequentially by running each expert one after another or in parallel by evaluating the experts concurrently. The profiling results below report per-patient rollout latency for one sampled future trajectory and training times on a single GH200 machine.

*Table 21*. **Per-patient rollout latency for one sampled future trajectory.** Sequential MoRGen runs the constituent experts one after another, while projected parallel MoRGen assumes the coarse experts are evaluated concurrently.

| Dataset | Baseline (ms) | Sequential MoRGen (ms) | Projected parallel MoRGen (ms) |
|---|---|---|---|
| EHRSHOT | 12.83 | 270.17 | 20.39 |
| MIMIC | 10.41 | 142.58 | 12.26 |

*Table 22*. **Training-time summary.** Baseline time is taken from the minute-level pretraining run. Coarse-expert time summarizes successful histogram-model runs, including both EM fitting and histogram autoregressive training for each $(\Delta, K)$ expert. The full sequential MoRGen cost sums the baseline and all $R$ coarse experts; in practice, these independent expert jobs can be parallelized.

| Dataset | Baseline (min) | Coarse expert median [min–max] | $R$ | Avg. added per expert (min) | Full sequential MoRGen cost (min) |
|---|---|---|---|---|---|
| EHRSHOT | 5.4 | 12.2 [10.6–15.6] | 20 | 13.0 | 265.4 |
| MIMIC | 8.4 | 30.8 [18.8–102.5] | 20 | 39.8 | 804.4 |

Table 22 reports both the marginal cost of adding a single coarse expert and the total sequential cost of training the full expert pool. The average added training cost per coarse expert is 13.0 minutes on EHRSHOT and 39.8 minutes on MIMIC, including both EM fitting and autoregressive training. Training all 20 coarse experts sequentially, together with the minute-level baseline, requires 265.4 minutes on EHRSHOT and 804.4 minutes on MIMIC. However, because the coarse experts are independent, this sequential total reflects total compute rather than an unavoidable wall-clock requirement; the jobs can be distributed across GPUs or machines.

At inference time, sequential MoRGen is substantially slower than the minute-level baseline, but the wall-clock cost decreases sharply when independent experts are parallelized. The projected parallel runtime reaches approximately $1.6\times$ baseline latency on EHRSHOT and $1.2\times$ baseline latency on MIMIC. Thus, MoRGen trades additional total computation for improved predictive performance, while remaining practical when independent experts are trained and evaluated in parallel.

## H. Raw Per-Model AUROC and BCE Tables

For completeness, we provide a unified raw-results appendix for the two headline horizons used in the main paper: 30 days and 2 years. For compactness, we present separate AUROC and BCE tables for each dataset, with the 30-day and 2-year task columns shown side by side within each table. These appendix tables report held-out point estimates for the minute-level baseline, every individual coarse expert in the sweep, and MoRGen itself; the main-text tables retain the corresponding jackknife standard errors for the headline baseline-versus-MoRGen comparisons.

*Table 23.* **MIMIC raw per-model held-out AUROC results.** Columns are split by horizon (30 days and 730 days / 2 years). Strategy abbreviations are: `Min` for the minute-level baseline, `D/W/Mo/Q/S` for day/week/month/quarter/semi-annual experts, and `MoR` for `MoRGen` convex fusion; for this dataset, numeric suffixes 2/4/8/16 denote $K \in \{2048, 4096, 8192, 16384\}$.

| Strategy | 30d | | | | | | | 730d | | | | | | |
| | Creat. | Death | Hct | Hgb | WBC | Plt | Readm. | Creat. | Death | Hct | Hgb | WBC | Plt | Readm. |
|---|---|---|---|---|---|---|---|---|---|---|---|---|---|---|
| Min | 92.16 | 70.27 | 88.21 | 88.15 | 85.96 | 93.91 | 69.98 | 90.19 | 59.22 | 82.30 | 82.71 | 83.03 | 85.18 | 68.32 |
| D2 | 90.75 | 84.52 | 86.81 | 86.33 | 80.98 | 90.72 | 63.51 | 82.09 | 76.42 | 81.30 | 82.09 | 75.51 | 79.44 | 67.04 |
| W2 | 88.13 | 80.67 | 84.54 | 84.30 | 81.58 | 89.90 | 65.48 | 82.50 | 76.18 | 80.12 | 80.90 | 76.36 | 80.81 | 64.88 |
| Mo2 | 88.37 | 75.78 | 82.41 | 82.53 | 79.64 | 88.63 | 66.33 | 83.99 | 77.70 | 79.27 | 80.33 | 75.74 | 80.46 | 66.91 |
| Q2 | 89.98 | 80.24 | 84.19 | 84.53 | 79.80 | 90.63 | 65.77 | 86.42 | 79.44 | 80.42 | 81.22 | 76.73 | 81.72 | 67.78 |
| S2 | 90.51 | 79.81 | 82.90 | 83.57 | 78.88 | 89.63 | 64.42 | 86.84 | 77.54 | 80.03 | 81.18 | 76.48 | 81.30 | 67.30 |
| D4 | 90.08 | 82.65 | 86.36 | 86.09 | 82.05 | 92.19 | 63.25 | 80.50 | 76.01 | 80.85 | 81.43 | 77.14 | 82.17 | 66.31 |
| W4 | 89.47 | 80.88 | 84.34 | 84.87 | 81.63 | 89.74 | 65.77 | 83.44 | 75.41 | 79.87 | 80.63 | 77.61 | 81.36 | 65.82 |
| Mo4 | 88.65 | 81.22 | 83.00 | 83.13 | 80.33 | 90.60 | 66.74 | 85.40 | 78.12 | 79.30 | 80.33 | 77.34 | 81.43 | 66.93 |
| Q4 | 90.40 | 80.00 | 84.26 | 84.48 | 80.44 | 89.80 | 65.99 | 85.73 | 79.41 | 80.26 | 81.17 | 77.75 | 81.58 | 67.69 |
| S4 | 90.16 | 80.20 | 82.38 | 82.82 | 78.66 | 88.54 | 64.13 | 86.77 | 78.86 | 79.83 | 81.08 | 76.60 | 81.41 | 67.13 |
| D8 | 89.71 | 83.41 | 85.73 | 86.04 | 81.22 | 91.97 | 62.41 | 79.84 | 73.50 | 80.39 | 81.31 | 76.06 | 80.92 | 66.09 |
| W8 | 87.60 | 82.58 | 81.98 | 82.56 | 81.16 | 90.53 | 65.58 | 81.66 | 77.04 | 77.90 | 79.12 | 78.29 | 80.93 | 64.65 |
| Mo8 | 88.17 | 81.33 | 80.90 | 81.12 | 78.65 | 89.56 | 65.18 | 85.36 | 79.15 | 78.17 | 78.98 | 76.54 | 80.57 | 67.24 |
| Q8 | 90.15 | 80.37 | 82.02 | 82.14 | 78.66 | 88.55 | 65.78 | 86.00 | 78.10 | 79.18 | 79.76 | 77.19 | 80.82 | 67.32 |
| S8 | 89.48 | 77.89 | 79.88 | 79.93 | 77.57 | 86.49 | 63.65 | 86.20 | 77.83 | 77.55 | 78.36 | 75.91 | 79.33 | 66.00 |
| D16 | 91.06 | 84.43 | 86.09 | 86.15 | 80.96 | 92.62 | 64.65 | 83.09 | 77.60 | 81.13 | 82.24 | 77.15 | 82.93 | 67.90 |
| W16 | 88.77 | 80.34 | 82.20 | 82.07 | 80.95 | 90.84 | 65.60 | 81.45 | 75.62 | 77.71 | 78.82 | 77.22 | 81.32 | 65.00 |
| Mo16 | 86.64 | 81.29 | 78.51 | 78.23 | 77.34 | 88.51 | 64.51 | 86.17 | 78.72 | 77.36 | 78.38 | 77.65 | 80.21 | 66.17 |
| Q16 | 88.18 | 80.17 | 79.67 | 79.54 | 77.35 | 87.57 | 64.55 | 84.62 | 77.19 | 77.92 | 78.76 | 76.73 | 79.92 | 66.56 |
| S16 | 89.29 | 79.23 | 81.32 | 81.11 | 77.31 | 88.83 | 64.42 | 85.69 | 76.75 | 78.97 | 79.66 | 76.22 | 79.90 | 66.27 |
| MoR | 93.34 | 89.06 | 89.39 | 89.42 | 86.31 | 94.67 | 70.28 | 90.77 | 81.75 | 83.45 | 84.04 | 83.31 | 85.82 | 70.05 |

*Table 24.* **MIMIC raw per-model held-out BCE results.** Columns are split by horizon (30 days and 730 days / 2 years). Strategy abbreviations are: `Min` for the minute-level baseline, `D/W/Mo/Q/S` for day/week/month/quarter/semi-annual experts, and `MoR` for `MoRGen` convex fusion; for this dataset, numeric suffixes 2/4/8/16 denote $K \in \{2048, 4096, 8192, 16384\}$.

| Strategy | 30d | | | | | | | 730d | | | | | | |
| --- | --- | --- | --- | --- | --- | --- | --- | --- | --- | --- | --- | --- | --- | --- |
| | Creat. | Death | Hct | Hgb | WBC | Plt | Readm. | Creat. | Death | Hct | Hgb | WBC | Plt | Readm. |
| Min | 0.168 | 0.478 | 0.228 | 0.227 | 0.381 | 0.144 | 0.561 | 0.368 | 1.928 | 0.484 | 0.486 | 0.608 | 0.356 | 0.966 |
| D2 | 0.140 | 0.162 | 0.269 | 0.289 | 0.333 | 0.114 | 0.645 | 0.413 | 0.637 | 0.614 | 0.651 | 0.593 | 0.373 | 0.825 |
| W2 | 0.136 | 0.162 | 0.375 | 0.380 | 0.347 | 0.116 | 1.092 | 0.313 | 0.557 | 0.998 | 1.043 | 0.571 | 0.277 | 1.293 |
| Mo2 | 0.173 | 0.158 | 0.421 | 0.406 | 0.597 | 0.188 | 1.571 | 0.319 | 0.444 | 1.096 | 1.062 | 0.618 | 0.303 | 1.449 |
| Q2 | 0.156 | 0.134 | 0.304 | 0.287 | 0.456 | 0.149 | 1.306 | 0.311 | 0.483 | 0.882 | 0.881 | 0.659 | 0.334 | 1.584 |
| S2 | 0.157 | 0.131 | 0.294 | 0.272 | 0.469 | 0.157 | 1.303 | 0.308 | 0.486 | 0.786 | 0.745 | 0.645 | 0.342 | 1.461 |
| D4 | 0.140 | 0.181 | 0.291 | 0.313 | 0.333 | 0.105 | 0.678 | 0.387 | 0.735 | 0.755 | 0.856 | 0.569 | 0.311 | 0.945 |
| W4 | 0.131 | 0.177 | 0.409 | 0.384 | 0.367 | 0.132 | 1.290 | 0.310 | 0.602 | 1.177 | 1.193 | 0.570 | 0.297 | 1.448 |
| Mo4 | 0.182 | 0.166 | 0.423 | 0.413 | 0.683 | 0.175 | 1.970 | 0.306 | 0.481 | 1.201 | 1.146 | 0.655 | 0.333 | 1.840 |
| Q4 | 0.160 | 0.150 | 0.331 | 0.318 | 0.478 | 0.156 | 1.482 | 0.331 | 0.502 | 0.977 | 0.967 | 0.662 | 0.364 | 1.978 |
| S4 | 0.165 | 0.130 | 0.309 | 0.295 | 0.492 | 0.180 | 1.242 | 0.354 | 0.451 | 0.809 | 0.767 | 0.672 | 0.356 | 1.509 |
| D8 | 0.133 | 0.140 | 0.339 | 0.320 | 0.334 | 0.104 | 0.645 | 0.402 | 0.478 | 0.799 | 0.803 | 0.578 | 0.322 | 0.799 |
| W8 | 0.148 | 0.154 | 0.512 | 0.486 | 0.399 | 0.131 | 1.462 | 0.316 | 0.519 | 1.334 | 1.323 | 0.583 | 0.288 | 1.491 |
| Mo8 | 0.202 | 0.153 | 0.544 | 0.530 | 0.917 | 0.204 | 2.303 | 0.317 | 0.427 | 1.380 | 1.365 | 0.656 | 0.341 | 1.787 |
| Q8 | 0.177 | 0.141 | 0.395 | 0.385 | 0.656 | 0.197 | 1.866 | 0.367 | 0.479 | 1.077 | 1.084 | 0.769 | 0.414 | 2.244 |
| S8 | 0.183 | 0.143 | 0.376 | 0.370 | 0.603 | 0.202 | 1.472 | 0.393 | 0.448 | 0.942 | 0.933 | 0.778 | 0.385 | 1.659 |
| D16 | 0.127 | 0.140 | 0.316 | 0.308 | 0.332 | 0.103 | 0.674 | 0.356 | 0.545 | 0.755 | 0.751 | 0.566 | 0.277 | 0.904 |
| W16 | 0.142 | 0.161 | 0.516 | 0.517 | 0.434 | 0.138 | 1.647 | 0.325 | 0.512 | 1.443 | 1.447 | 0.643 | 0.308 | 1.715 |
| Mo16 | 0.255 | 0.170 | 0.638 | 0.631 | 1.066 | 0.233 | 2.767 | 0.313 | 0.427 | 1.490 | 1.415 | 0.698 | 0.365 | 1.934 |
| Q16 | 0.213 | 0.144 | 0.446 | 0.444 | 0.766 | 0.215 | 1.927 | 0.430 | 0.457 | 1.198 | 1.185 | 0.831 | 0.461 | 2.041 |
| S16 | 0.203 | 0.132 | 0.369 | 0.362 | 0.672 | 0.186 | 1.569 | 0.444 | 0.449 | 0.922 | 0.915 | 0.849 | 0.437 | 1.814 |
| MoR | 0.101 | 0.103 | 0.165 | 0.167 | 0.293 | 0.080 | 0.548 | 0.237 | 0.407 | 0.373 | 0.372 | 0.484 | 0.228 | 0.593 |

*Table 25.* **Hospital A raw per-model held-out AUROC results.** Columns are split by horizon (30 days and 730 days / 2 years). Strategy abbreviations are: `Min` for the minute-level baseline, `D/W/Mo/Q/S` for day/week/month/quarter/semi-annual experts, and `MoR` for `MoRGen` convex fusion; for this dataset, numeric suffixes 2/4/8/16 denote $K \in \{2048, 4096, 8192, 16384\}$.

| Strategy | 30d | | | | | | | 730d | | | | | | |
| --- | --- | --- | --- | --- | --- | --- | --- | --- | --- | --- | --- | --- | --- | --- |
| | Creat. | Death | Hct | Hgb | WBC | Plt | Readm. | Creat. | Death | Hct | Hgb | WBC | Plt | Readm. |
| Min | 94.50 | 69.13 | 91.29 | 91.63 | 87.39 | 93.49 | 85.11 | 89.40 | 58.93 | 82.08 | 83.01 | 80.60 | 84.69 | 73.38 |
| D2 | 87.92 | 78.54 | 90.59 | 90.86 | 80.95 | 90.79 | 82.29 | 81.72 | 73.96 | 82.61 | 83.27 | 74.37 | 80.47 | 74.94 |
| W2 | 91.65 | 76.76 | 87.51 | 87.85 | 82.81 | 89.78 | 76.41 | 86.44 | 73.15 | 80.69 | 81.52 | 76.38 | 81.19 | 68.39 |
| Mo2 | 90.83 | 75.66 | 85.87 | 85.99 | 81.43 | 88.07 | 71.55 | 86.62 | 73.88 | 80.34 | 80.93 | 76.30 | 80.69 | 67.69 |
| Q2 | 90.27 | 74.94 | 83.68 | 83.85 | 79.54 | 85.58 | 66.15 | 86.96 | 72.86 | 79.77 | 80.42 | 76.01 | 79.11 | 66.68 |
| S2 | 88.34 | 71.96 | 80.62 | 81.01 | 76.15 | 83.39 | 57.96 | 86.13 | 70.89 | 78.22 | 78.83 | 74.88 | 78.39 | 64.68 |
| D4 | 89.43 | 78.40 | 90.69 | 90.89 | 81.64 | 91.41 | 82.34 | 83.45 | 73.33 | 82.51 | 83.04 | 74.92 | 81.06 | 74.96 |
| W4 | 91.97 | 76.91 | 87.62 | 88.04 | 83.19 | 90.28 | 76.37 | 86.75 | 73.86 | 80.71 | 81.61 | 77.38 | 81.83 | 68.54 |
| Mo4 | 91.23 | 75.79 | 85.61 | 85.70 | 81.59 | 88.23 | 70.91 | 86.89 | 73.17 | 80.23 | 80.75 | 76.32 | 79.98 | 66.84 |
| Q4 | 89.95 | 74.98 | 83.23 | 83.55 | 80.11 | 85.86 | 65.95 | 86.89 | 73.06 | 79.57 | 80.34 | 76.64 | 79.69 | 67.06 |
| S4 | 89.65 | 73.17 | 82.22 | 82.29 | 78.03 | 84.86 | 61.74 | 87.19 | 72.05 | 79.23 | 79.72 | 76.82 | 80.06 | 66.21 |
| D8 | 91.63 | 76.49 | 90.60 | 90.76 | 83.03 | 92.22 | 81.13 | 85.32 | 72.32 | 82.79 | 83.22 | 76.39 | 81.85 | 73.96 |
| W8 | 92.54 | 76.83 | 87.58 | 88.12 | 83.39 | 90.36 | 76.12 | 87.15 | 72.99 | 80.62 | 81.60 | 77.58 | 81.85 | 68.25 |
| Mo8 | 91.24 | 75.11 | 85.31 | 85.44 | 81.65 | 88.14 | 71.75 | 86.81 | 73.30 | 79.89 | 80.43 | 76.73 | 79.98 | 67.01 |
| Q8 | 89.82 | 73.74 | 82.11 | 82.32 | 79.31 | 85.74 | 64.66 | 86.30 | 71.90 | 78.79 | 79.47 | 76.55 | 79.86 | 66.56 |
| S8 | 90.10 | 73.61 | 81.79 | 81.87 | 77.59 | 85.26 | 61.67 | 87.72 | 72.39 | 78.71 | 79.19 | 76.41 | 80.28 | 66.37 |
| D16 | 92.59 | 77.11 | 90.43 | 90.65 | 83.59 | 92.09 | 81.20 | 86.63 | 73.10 | 82.36 | 82.92 | 76.62 | 81.70 | 74.36 |
| W16 | 92.58 | 76.58 | 87.38 | 87.75 | 83.66 | 90.37 | 75.68 | 87.27 | 73.00 | 80.26 | 81.15 | 78.26 | 82.01 | 67.23 |
| Mo16 | 91.39 | 75.42 | 85.20 | 85.17 | 82.06 | 88.34 | 72.31 | 87.25 | 73.79 | 79.89 | 80.44 | 78.06 | 81.19 | 67.91 |
| Q16 | 90.27 | 72.56 | 82.37 | 82.58 | 79.81 | 86.11 | 65.96 | 87.21 | 73.03 | 79.02 | 79.84 | 77.10 | 80.14 | 67.08 |
| S16 | 89.59 | 71.70 | 80.22 | 80.40 | 77.49 | 83.85 | 60.76 | 86.86 | 71.98 | 77.71 | 78.35 | 76.20 | 78.75 | 65.77 |
| MoR | 95.29 | 81.67 | 91.80 | 92.13 | 87.73 | 93.94 | 85.50 | 90.41 | 77.46 | 83.85 | 84.66 | 81.81 | 85.45 | 77.46 |

*Table 26.* **Hospital A raw per-model held-out BCE results.** Columns are split by horizon (30 days and 730 days / 2 years). Strategy abbreviations are: `Min` for the minute-level baseline, `D/W/Mo/Q/S` for day/week/month/quarter/semi-annual experts, and `MoR` for `MoRGen` convex fusion; for this dataset, numeric suffixes 2/4/8/16 denote $K \in \{2048, 4096, 8192, 16384\}$.

| Strategy | 30d | | | | | | | 730d | | | | | | |
|---|---|---|---|---|---|---|---|---|---|---|---|---|---|---|
| | Creat. | Death | Hct | Hgb | WBC | Plt | Readm. | Creat. | Death | Hct | Hgb | WBC | Plt | Readm. |
| Min | 0.261 | 0.606 | 0.336 | 0.317 | 0.400 | 0.237 | 0.479 | 0.396 | 1.271 | 0.580 | 0.556 | 0.661 | 0.418 | 0.661 |
| D2 | 0.414 | 0.207 | 0.396 | 0.387 | 0.537 | 0.296 | 0.518 | 0.737 | 0.523 | 0.610 | 0.616 | 0.723 | 0.577 | 0.487 |
| W2 | 0.232 | 0.225 | 0.485 | 0.466 | 0.421 | 0.246 | 0.722 | 0.359 | 0.539 | 0.887 | 0.858 | 0.587 | 0.408 | 0.616 |
| Mo2 | 0.248 | 0.214 | 0.466 | 0.451 | 0.452 | 0.279 | 0.789 | 0.345 | 0.547 | 0.922 | 0.908 | 0.594 | 0.443 | 0.702 |
| Q2 | 0.324 | 0.221 | 0.555 | 0.536 | 0.540 | 0.365 | 1.001 | 0.351 | 0.561 | 0.840 | 0.822 | 0.607 | 0.462 | 0.718 |
| S2 | 0.410 | 0.237 | 0.671 | 0.634 | 0.684 | 0.450 | 1.280 | 0.388 | 0.580 | 0.798 | 0.771 | 0.642 | 0.489 | 0.794 |
| D4 | 0.383 | 0.207 | 0.419 | 0.409 | 0.516 | 0.272 | 0.525 | 0.679 | 0.529 | 0.690 | 0.711 | 0.703 | 0.536 | 0.492 |
| W4 | 0.226 | 0.215 | 0.487 | 0.471 | 0.418 | 0.240 | 0.708 | 0.350 | 0.555 | 0.980 | 0.965 | 0.576 | 0.407 | 0.616 |
| Mo4 | 0.255 | 0.227 | 0.541 | 0.521 | 0.477 | 0.293 | 1.075 | 0.349 | 0.557 | 1.182 | 1.149 | 0.629 | 0.479 | 0.886 |
| Q4 | 0.323 | 0.224 | 0.592 | 0.563 | 0.539 | 0.365 | 1.072 | 0.360 | 0.562 | 0.928 | 0.892 | 0.606 | 0.473 | 0.790 |
| S4 | 0.375 | 0.232 | 0.642 | 0.619 | 0.643 | 0.425 | 1.211 | 0.367 | 0.560 | 0.832 | 0.808 | 0.617 | 0.476 | 0.746 |
| D8 | 0.309 | 0.210 | 0.414 | 0.420 | 0.483 | 0.244 | 0.539 | 0.572 | 0.543 | 0.677 | 0.738 | 0.656 | 0.489 | 0.508 |
| W8 | 0.219 | 0.221 | 0.563 | 0.522 | 0.425 | 0.247 | 0.772 | 0.330 | 0.538 | 1.190 | 1.143 | 0.584 | 0.424 | 0.675 |
| Mo8 | 0.249 | 0.237 | 0.567 | 0.538 | 0.500 | 0.310 | 1.117 | 0.347 | 0.563 | 1.235 | 1.200 | 0.650 | 0.495 | 0.858 |
| Q8 | 0.348 | 0.232 | 0.664 | 0.631 | 0.593 | 0.386 | 1.329 | 0.411 | 0.592 | 1.135 | 1.098 | 0.681 | 0.529 | 0.990 |
| S8 | 0.363 | 0.228 | 0.647 | 0.616 | 0.646 | 0.416 | 1.180 | 0.368 | 0.548 | 0.870 | 0.839 | 0.636 | 0.482 | 0.731 |
| D16 | 0.265 | 0.211 | 0.483 | 0.472 | 0.451 | 0.226 | 0.530 | 0.486 | 0.546 | 0.933 | 1.006 | 0.631 | 0.459 | 0.499 |
| W16 | 0.222 | 0.221 | 0.565 | 0.561 | 0.440 | 0.275 | 0.773 | 0.342 | 0.560 | 1.236 | 1.312 | 0.602 | 0.444 | 0.697 |
| Mo16 | 0.260 | 0.240 | 0.566 | 0.542 | 0.521 | 0.313 | 1.092 | 0.368 | 0.573 | 1.225 | 1.162 | 0.637 | 0.503 | 0.841 |
| Q16 | 0.328 | 0.226 | 0.656 | 0.632 | 0.575 | 0.380 | 1.332 | 0.389 | 0.565 | 1.109 | 1.069 | 0.665 | 0.525 | 0.926 |
| S16 | 0.385 | 0.236 | 0.749 | 0.715 | 0.662 | 0.458 | 1.476 | 0.399 | 0.570 | 1.018 | 0.987 | 0.678 | 0.543 | 0.990 |
| MoR | 0.160 | 0.185 | 0.280 | 0.271 | 0.355 | 0.179 | 0.459 | 0.284 | 0.499 | 0.456 | 0.445 | 0.519 | 0.358 | 0.421 |

*Table 27.* **EHRSHOT raw per-model held-out AUROC results.** Columns are split by horizon (30 days and 730 days / 2 years). Strategy abbreviations are: `Min` for the minute-level baseline, `D/W/Mo/Q/S` for day/week/month/quarter/semi-annual experts, and `MoR` for `MoRGen` convex fusion; for this dataset, the numeric suffix is the exact $K$ value.

| Strategy | 30d | | | | | | | 730d | | | | | | |
|---|---|---|---|---|---|---|---|---|---|---|---|---|---|---|
| | Creat. | Death | Hct | Hgb | WBC | Plt | Readm. | Creat. | Death | Hct | Hgb | WBC | Plt | Readm. |
| Min | 77.85 | 49.98 | 79.08 | 76.93 | 66.20 | 83.60 | 63.86 | 82.23 | 49.98 | 78.71 | 77.23 | 62.71 | 79.96 | 65.72 |
| D64 | 71.35 | 73.99 | 82.04 | 81.25 | 69.79 | 84.81 | 69.24 | 75.09 | 62.17 | 76.78 | 76.35 | 67.29 | 77.81 | 67.97 |
| W64 | 76.86 | 74.64 | 84.06 | 84.18 | 75.76 | 90.04 | 66.24 | 78.07 | 71.49 | 79.34 | 78.95 | 68.02 | 80.67 | 68.13 |
| Mo64 | 70.42 | 76.16 | 81.85 | 81.79 | 78.77 | 87.51 | 68.26 | 76.32 | 72.57 | 77.21 | 77.42 | 68.76 | 78.99 | 69.12 |
| Q64 | 69.67 | 76.65 | 79.57 | 77.67 | 71.91 | 80.57 | 62.53 | 74.20 | 71.69 | 74.18 | 73.40 | 67.08 | 73.59 | 65.47 |
| S64 | 76.42 | 71.41 | 76.59 | 74.57 | 73.38 | 78.33 | 60.91 | 75.37 | 73.20 | 73.22 | 73.11 | 70.02 | 73.79 | 65.29 |
| D128 | 70.14 | 78.95 | 83.73 | 83.09 | 71.26 | 87.66 | 69.64 | 71.86 | 68.81 | 78.65 | 77.97 | 65.86 | 79.25 | 70.68 |
| W128 | 79.03 | 67.92 | 82.11 | 82.68 | 72.70 | 89.63 | 67.21 | 78.08 | 67.78 | 78.07 | 77.91 | 66.58 | 81.23 | 67.57 |
| Mo128 | 80.16 | 79.38 | 82.78 | 82.44 | 73.22 | 88.04 | 67.57 | 79.76 | 74.30 | 77.75 | 77.46 | 67.22 | 80.02 | 68.04 |
| Q128 | 72.06 | 79.12 | 81.11 | 81.02 | 74.53 | 85.69 | 64.46 | 75.19 | 73.22 | 76.08 | 76.03 | 68.87 | 78.44 | 66.18 |
| S128 | 76.78 | 77.36 | 80.25 | 79.26 | 75.96 | 81.99 | 63.67 | 78.36 | 73.88 | 75.74 | 75.65 | 71.33 | 75.94 | 67.29 |
| D256 | 76.95 | 80.92 | 84.04 | 83.85 | 76.74 | 88.94 | 68.58 | 73.01 | 67.66 | 79.25 | 79.02 | 65.28 | 80.84 | 69.34 |
| W256 | 78.57 | 78.97 | 81.54 | 80.31 | 69.57 | 88.98 | 64.53 | 78.95 | 72.29 | 76.27 | 75.87 | 63.60 | 80.16 | 65.88 |
| Mo256 | 76.89 | 77.42 | 80.60 | 79.16 | 72.45 | 88.03 | 66.15 | 79.64 | 72.04 | 75.95 | 75.70 | 66.08 | 79.62 | 67.96 |
| Q256 | 76.77 | 72.74 | 77.75 | 78.60 | 79.13 | 84.63 | 61.92 | 76.92 | 71.56 | 74.20 | 73.43 | 70.77 | 77.25 | 66.49 |
| S256 | 80.55 | 74.99 | 75.61 | 74.75 | 73.15 | 80.36 | 60.46 | 73.19 | 72.27 | 72.79 | 73.58 | 67.43 | 75.00 | 64.84 |
| D512 | 79.82 | 77.76 | 82.85 | 82.74 | 70.92 | 87.70 | 68.37 | 76.44 | 65.54 | 79.21 | 78.81 | 64.33 | 80.27 | 70.00 |
| W512 | 82.68 | 80.31 | 83.38 | 83.69 | 69.45 | 91.08 | 67.37 | 81.72 | 73.25 | 78.28 | 77.92 | 66.09 | 82.10 | 69.09 |
| Mo512 | 78.57 | 79.91 | 79.91 | 79.31 | 74.07 | 87.11 | 65.69 | 78.04 | 75.55 | 76.17 | 74.97 | 65.07 | 81.32 | 66.37 |
| Q512 | 74.38 | 70.82 | 76.87 | 77.27 | 64.67 | 83.00 | 62.75 | 76.13 | 68.17 | 73.90 | 73.85 | 62.77 | 76.04 | 65.66 |
| S512 | 75.08 | 70.89 | 70.38 | 69.54 | 70.46 | 73.29 | 58.44 | 73.38 | 68.91 | 70.32 | 69.90 | 66.70 | 71.47 | 66.37 |
| MoR | 84.68 | 81.46 | 86.54 | 86.06 | 81.06 | 92.21 | 72.35 | 86.30 | 77.57 | 83.71 | 82.86 | 73.86 | 86.53 | 73.49 |

*Table 28.* **EHRSHOT raw per-model held-out BCE results.** Columns are split by horizon (30 days and 730 days / 2 years). Strategy abbreviations are: `Min` for the minute-level baseline, `D/W/Mo/Q/S` for day/week/month/quarter/semi-annual experts, and `MoR` for `MoRGen` convex fusion; for this dataset, the numeric suffix is the exact $K$ value.

| Strategy | 30d | | | | | | | 730d | | | | | | |
| | Creat. | Death | Hct | Hgb | WBC | Plt | Readm. | Creat. | Death | Hct | Hgb | WBC | Plt | Readm. |
|---|---|---|---|---|---|---|---|---|---|---|---|---|---|---|
| Min | 0.210 | 0.462 | 0.320 | 0.317 | 0.433 | 0.324 | 0.619 | 0.324 | 3.571 | 0.709 | 0.669 | 0.753 | 0.578 | 1.178 |
| D64 | 0.229 | 0.114 | 0.243 | 0.223 | 0.266 | 0.275 | 0.413 | 0.553 | 0.481 | 0.467 | 0.453 | 0.740 | 0.516 | 0.654 |
| W64 | 0.142 | 0.082 | 0.232 | 0.200 | 0.265 | 0.249 | 0.424 | 0.389 | 0.393 | 0.437 | 0.427 | 0.712 | 0.509 | 0.643 |
| Mo64 | 0.130 | 0.080 | 0.235 | 0.218 | 0.221 | 0.263 | 0.433 | 0.305 | 0.392 | 0.474 | 0.434 | 0.547 | 0.450 | 0.649 |
| Q64 | 0.129 | 0.080 | 0.246 | 0.214 | 0.226 | 0.321 | 0.470 | 0.282 | 0.388 | 0.486 | 0.470 | 0.490 | 0.476 | 0.680 |
| S64 | 0.135 | 0.093 | 0.314 | 0.266 | 0.238 | 0.405 | 0.549 | 0.266 | 0.424 | 0.512 | 0.487 | 0.475 | 0.472 | 0.685 |
| D128 | 0.286 | 0.134 | 0.273 | 0.264 | 0.299 | 0.281 | 0.427 | 0.929 | 0.626 | 0.620 | 0.623 | 0.925 | 0.671 | 0.738 |
| W128 | 0.135 | 0.104 | 0.255 | 0.198 | 0.250 | 0.234 | 0.422 | 0.333 | 0.434 | 0.459 | 0.462 | 0.629 | 0.445 | 0.669 |
| Mo128 | 0.127 | 0.086 | 0.256 | 0.229 | 0.227 | 0.247 | 0.452 | 0.263 | 0.402 | 0.508 | 0.477 | 0.526 | 0.429 | 0.693 |
| Q128 | 0.128 | 0.089 | 0.285 | 0.235 | 0.221 | 0.315 | 0.514 | 0.268 | 0.389 | 0.532 | 0.518 | 0.458 | 0.441 | 0.743 |
| S128 | 0.130 | 0.085 | 0.286 | 0.242 | 0.235 | 0.374 | 0.540 | 0.259 | 0.391 | 0.488 | 0.468 | 0.445 | 0.451 | 0.698 |
| D256 | 0.289 | 0.160 | 0.259 | 0.246 | 0.297 | 0.257 | 0.436 | 0.990 | 0.692 | 0.523 | 0.542 | 1.028 | 0.641 | 0.713 |
| W256 | 0.138 | 0.076 | 0.265 | 0.258 | 0.276 | 0.237 | 0.479 | 0.307 | 0.425 | 0.538 | 0.533 | 0.601 | 0.438 | 0.710 |
| Mo256 | 0.130 | 0.091 | 0.326 | 0.289 | 0.284 | 0.264 | 0.583 | 0.256 | 0.444 | 0.796 | 0.709 | 0.501 | 0.471 | 0.809 |
| Q256 | 0.125 | 0.083 | 0.285 | 0.239 | 0.211 | 0.311 | 0.514 | 0.257 | 0.404 | 0.555 | 0.539 | 0.465 | 0.449 | 0.751 |
| S256 | 0.134 | 0.091 | 0.323 | 0.263 | 0.256 | 0.409 | 0.598 | 0.268 | 0.427 | 0.566 | 0.515 | 0.444 | 0.479 | 0.776 |
| D512 | 0.233 | 0.128 | 0.277 | 0.259 | 0.285 | 0.256 | 0.438 | 0.704 | 0.566 | 0.479 | 0.489 | 0.768 | 0.544 | 0.670 |
| W512 | 0.119 | 0.086 | 0.324 | 0.254 | 0.307 | 0.220 | 0.507 | 0.270 | 0.441 | 0.757 | 0.730 | 0.608 | 0.426 | 0.775 |
| Mo512 | 0.119 | 0.075 | 0.297 | 0.268 | 0.282 | 0.271 | 0.527 | 0.276 | 0.425 | 0.702 | 0.764 | 0.559 | 0.478 | 0.838 |
| Q512 | 0.125 | 0.083 | 0.293 | 0.242 | 0.270 | 0.325 | 0.561 | 0.287 | 0.405 | 0.601 | 0.575 | 0.508 | 0.502 | 0.860 |
| S512 | 0.131 | 0.087 | 0.314 | 0.258 | 0.263 | 0.391 | 0.627 | 0.271 | 0.401 | 0.567 | 0.529 | 0.469 | 0.504 | 0.788 |
| MoR | 0.110 | 0.076 | 0.206 | 0.186 | 0.201 | 0.204 | 0.397 | 0.227 | 0.347 | 0.401 | 0.393 | 0.421 | 0.365 | 0.601 |

# I. Mixed-Timescale Sequence Fusion

We also explored a mixed-timescale sequence model that interleaves coarse window-level tokens with fine-grained observation tokens in a single autoregressive sequence. In this variant, the model first generates a coarse month-level binary-presence token summarizing the next window, and then generates the minute-level observation sequence within that month conditioned on the generated window token. The motivation was to combine the stability of coarse temporal abstraction with the higher-resolution detail of the minute-level event stream.

In preliminary experiments, this sequence-fusion approach did not improve performance reliably relative to the minute-level baseline. Although coarse fixed-window models were sometimes competitive for terminal or end-of-timeline events, the mixed-timescale sequence model consistently underperformed on the broader set of intermittent outcomes used in our main evaluation, including readmission and abnormal laboratory events. We therefore did not include this model in the main MoRGen formulation. Instead, MoRGen keeps resolution-specific experts separate and fuses their endpoint-level risk estimates, which avoids forcing a single autoregressive decoder to jointly model coarse window summaries and the fine-grained event stream within each window.

*Table 29.* **Cohort sizes and data splits.** Splits are by patient ID for all datasets; we use the official EHRSHOT splits.

| Dataset | # Patients | # Discharges | Train/Val/Test (%) |
|---|---|---|---|
| Hospital A | 191,691 | 1,531,893 | 95/2.5/2.5 |
| MIMIC | 154,269 | 397,737 | 95/2.5/2.5 |
| EHRSHOT | 5,717 | 7,256 | 33/33/33 |

*Table 30.* **Held-out cohort characteristics and task prevalences.** Age is years at indexed discharge; follow-up is observed post-discharge follow-up in days. The table records the median and inter-quartile range (IQR) for age and follow-up. Task prevalence is the percentage of held-out indexed discharges with the outcome observed within the stated horizon.

| Metric | MIMIC | Hospital A | EHRSHOT |
|---|---|---|---|
| Age at discharge (years) [IQR] | 62.6 [51.1, 75.0] | 69.9 [59.4, 79.3] | 57.5 [43.7, 68.1] |
| Follow-up (days) [IQR] | 644.3 [196.1, 1772.0] | 1008.8 [214.0, 2587.0] | 843.5 [255.0, 2086.6] |
| Creat. 30d prevalence | 4.4 | 13.0 | 3.3 |
| Creat. 730d prevalence | 14.3 | 21.5 | 9.7 |
| Death 30d prevalence | 3.3 | 5.8 | 1.8 |
| Death 730d prevalence | 21.3 | 28.0 | 14.1 |
| Hct 30d prevalence | 6.4 | 20.2 | 8.7 |
| Hct 730d prevalence | 19.2 | 33.7 | 24.7 |
| Hgb 30d prevalence | 6.3 | 19.4 | 7.1 |
| Hgb 730d prevalence | 19.7 | 33.3 | 22.9 |
| WBC 30d prevalence | 15.2 | 24.3 | 7.1 |
| WBC 730d prevalence | 39.0 | 45.9 | 19.6 |
| Plt 30d prevalence | 4.2 | 13.0 | 12.5 |
| Plt 730d prevalence | 10.4 | 22.4 | 24.4 |
| Readm. 30d prevalence | 28.9 | 47.8 | 17.5 |
| Readm. 730d prevalence | 68.7 | 80.1 | 48.8 |

## J. Dataset Details

**Datasets, tasks, and cohort construction.** We evaluate on three datasets: Hospital A (a heart failure cohort from a large tertiary care center), MIMIC-IV (Johnson et al., 2024), and EHRSHOT (Wornow et al., 2023). We report full cohort sizes and split proportions in Table 29, held-out cohort characteristics and task prevalences at the headline horizons in Table 30, and outcome prevalence trends across horizons in Figure 5. For Hospital A and MIMIC, we split by patient IDs; for EHRSHOT, we use the provided patient splits.

To keep zero-shot generation tractable for the fine-grained baseline, we apply aggressive vocabulary filtering and retain only the events needed for our evaluated outcomes and trajectory boundaries: hospital admissions and discharges; leukocyte, platelet, hemoglobin, hematocrit, and creatinine labs; death and birth; and a special "timeline end" token. This filtering is applied for all models, but primarily helps reduce baseline context lengths, which has a large inference cost. On EHRSHOT we include a small number of additional codes that are low frequency: gender, race, care site, ethnicity, and diagnoses for acute myocardial infarction, lupus, celiac, pancreatic cancer, hyperlipidemia, and hypertension.

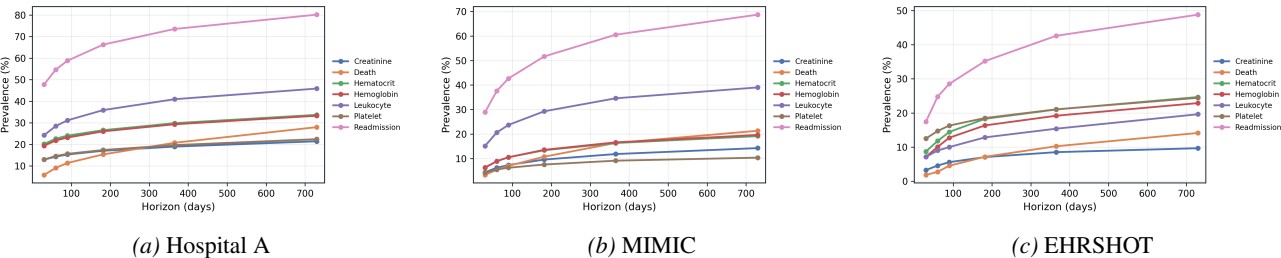

*(a)* Hospital A  *(b)* MIMIC  *(c)* EHRSHOT

*Figure 5.* **Outcome prevalence versus horizon.** Task prevalence (probability) across horizons (30 days to 2 years) for each dataset.

## K. Implementation Details

**Context length and rollout protocol.** For fairness, we cap *all* models (fine-grained and coarse) to a maximum context length of 512 tokens. Considering the low number of included observations this is a reasonable amount of context. In practice, coarse resolutions typically require far fewer than 512 tokens to represent post-discharge trajectories, so truncation is rare for coarse models.

At inference time, we condition on up to 128 tokens of pre-discharge history and generate up to 384 tokens post-discharge (i.e., a 512-token total budget). For the fine-grained baseline, this corresponds to conditioning on the most recent observed events prior to discharge (up to the token budget). For coarse models, the same budget applies to the coarsened sequence representation described below.

**Baseline pretraining.** The baseline model is trained with teacher forcing on next-token prediction. During training, we sample a contiguous subsequence of up to 512 tokens from each patient trajectory (with standard random cropping when trajectories exceed the budget).

**Coarse-resolution preprocessing and Bernoulli-mixture codebooks.** Each coarse resolution is defined by a fixed window size $\Delta$ (day/week/month/quarter/semi-annual). For a chosen $\Delta$, we convert trajectories into sequences of window-level *presence vectors* over the (filtered) vocabulary and fit a Bernoulli mixture model with $K$ components to compress each presence vector into a discrete window token (mixture assignment). We fit the Bernoulli mixture model using EM (Bishop, 2016).

**Choosing $K$.** We sweep $K$ by reconstruction performance (macro AUROC for reconstructing the window presence vectors). Across resolutions, the best-performing $K$ values concentrate in a small range, so we fix four $K$ values per dataset and use them throughout our coarse-model and MoRGen experiments. For Hospital A and MIMIC, we use $K \in \{2048, 4096, 8192, 16384\}$; for EHRSHOT, we use $K \in \{64, 128, 256, 512\}$.

**Initialization and selection.** The Bernoulli mixture model is sensitive to initialization; we found k-means++ initialization (Arthur & Vassilvitskii, 2007) consistently better than random initialization. For each $K$, we run 3 random seeds, perform k-means++ initialization, run 5 EM iterations, and select the (seed, iteration) checkpoint with the highest validation macro AUROC.

**Anchoring windows to discharge.** Presence vectors (and thus window tokens) are computed relative to a discharge time. Concretely, windows are aligned so that at least one window ends at the discharge (the "discharge window"), ensuring consistent temporal alignment across patients and resolutions (e.g., month-level windows correspond to contiguous 30-day blocks stepping backward/forward from discharge).

**Coarse-resolution token sequences.** Given a fitted Bernoulli mixture model at resolution $\Delta$, we convert each window to its discrete mixture assignment token. We also include time-gap tokens for runs of empty windows: if there is a gap of $R$ consecutive windows with no observed events, we replace them with a single gap token encoding $R$ (in units of windows). We append an EOS token to each sequence, and insert an explicit ANCHOR token immediately after the discharge window to mark the discharge position at inference time.

During coarse-model training, we sample a random patient and a random discharge, create a windowed sequence anchored at that discharge, convert windows to discrete tokens, and apply random cropping to 512 tokens if needed (just as we do with the baseline). During inference, we condition on up to 128 tokens pre-discharge and generate forward. For each generated

window token, we then sample a binary presence vector from the corresponding Bernoulli distribution (i.e., the mixture component's parameters) to obtain within-window event realizations.

**Typical generation lengths by resolution.**    Under our rollout horizons and alignment, coarse models require substantially shorter generations than the minute-level baseline. In our setup, all task horizons are at a maximum of 2 years, so we set post-discharge generation lengths to: semi-annual: 6 tokens; quarter: 12; month: 26; week: 128; day: 382 (plus special tokens such as EOS and ANCHOR, and any time-gap tokens for empty-window runs). We also set the max input tokens during inference to be 128 like for the baseline.

**Interpolating coarse-resolution forecasts.**    To evaluate each generative expert at a common forecasting horizon $H$, we first convert its sampled trajectories into an empirical survival curve $S(t) = \Pr(T > t)$, estimated as the fraction of samples whose event time exceeds $t$. For the minute-level baseline, the risk at horizon $H$ is computed directly as $1 - S(H)$. For a coarse expert with resolution $\Delta$, we evaluate survival at the enclosing grid points $t_{\text{low}} = \lfloor H/\Delta \rfloor \Delta$ and $t_{\text{high}} = t_{\text{low}} + \Delta$, and estimate

$$S(H) = \exp\left( \log S(t_{\text{low}}) + \frac{H - t_{\text{low}}}{\Delta} \left[\log S(t_{\text{high}}) - \log S(t_{\text{low}})\right] \right).$$

This log-linear interpolation is equivalent to assuming a piecewise-constant hazard within each coarse interval, and yields the forecasted risk $1 - S(H)$. Thus, coarse experts can still be scored at fine-grained horizons; for example, a semi-annual expert can produce a 30-day risk estimate by interpolating between its survival estimates at days 0 and 182.

**Model architecture and optimization.**    All experiments use a light-weight decoder-only Transformer with 4 layers, 4 attention heads, head dimension 64 (hidden size 256), and intermediate size 1024. We train with AdamW (Loshchilov & Hutter, 2019) using learning rate $1 \times 10^{-3}$ and weight decay 0.01. These hyperparameters performed best for the baseline across datasets, and we reuse them for the coarse-resolution models.

# L. Horizon Vs Performance Plots

This section includes BCE plots over varying horizons.

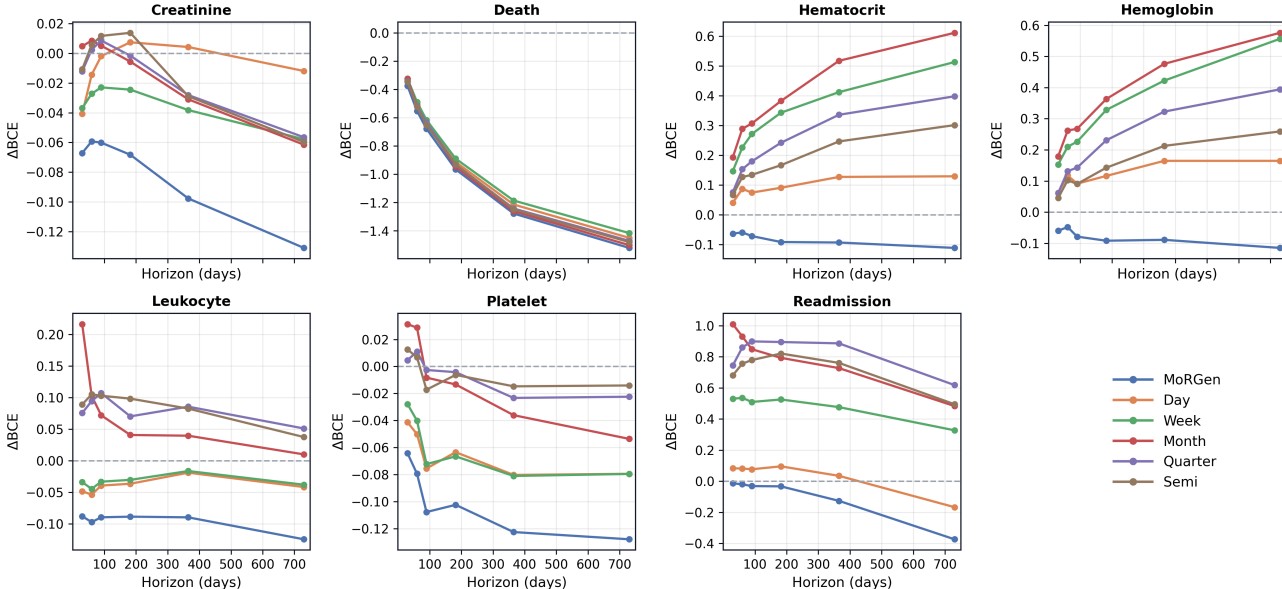

*Figure 6.* **MIMIC: BCE vs. horizon (days) across tasks.**

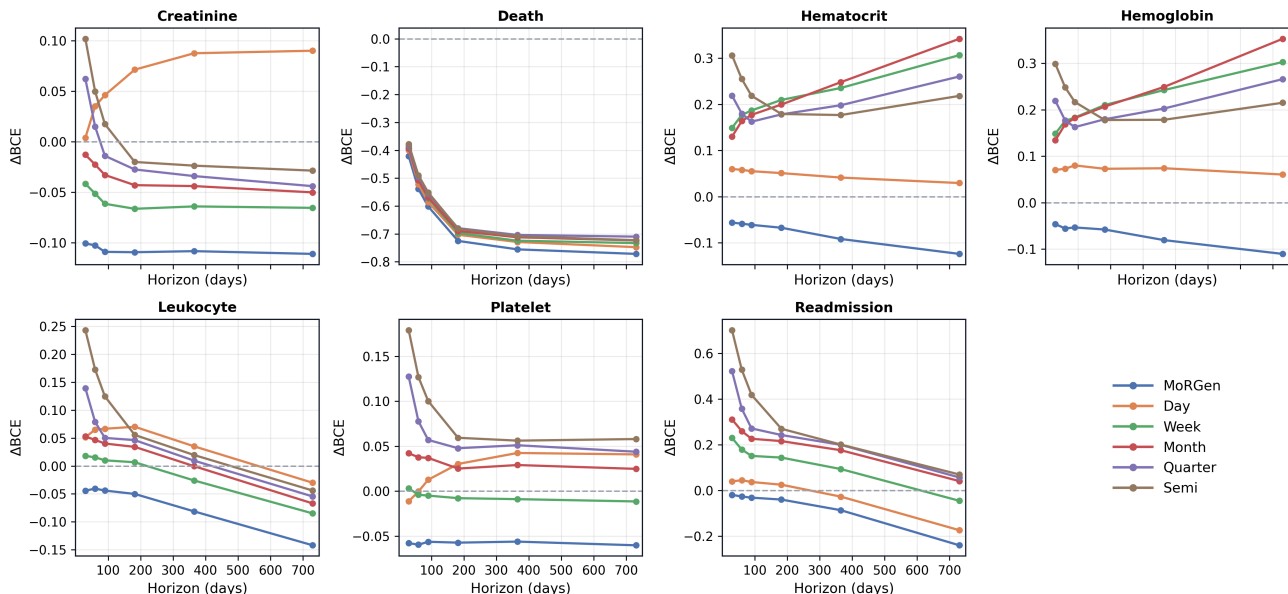

*Figure 7.* **Hospital A: BCE vs. horizon (days) across tasks.**

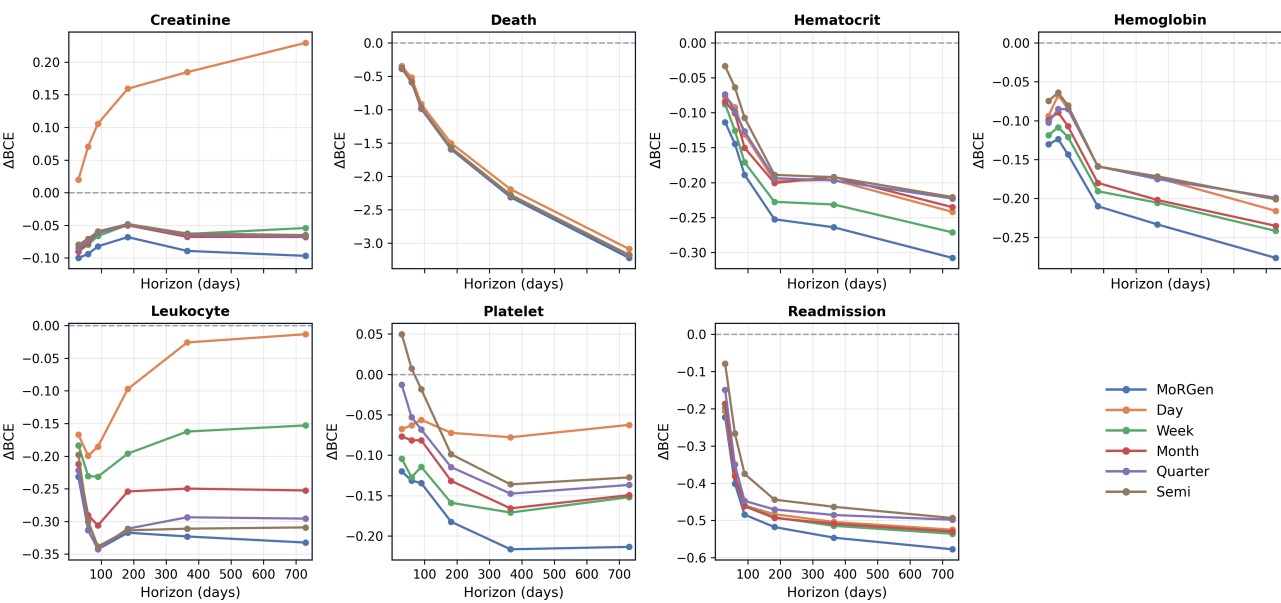

*Figure 8.* **EHRSHOT: BCE vs. horizon (days) across tasks.**

