# OpenReview forum: "MoRGen: Mixture-of-Resolutions Generative Forecasting for Irregularly Sampled Medical Time-Series Data"
_ICML.cc/2026/Conference — ICML 2026 regular_

### Official Review · Reviewer_aUJc · 2026-03-04

**Soundness:** 3
**Presentation:** 3
**Significance:** 3
**Originality:** 3
**Overall Recommendation:** 4
**Confidence:** 3

**Summary:**

This paper proposes MoRGen, a mixture-of-resolutions framework for zero-shot generative forecasting on irregularly sampled EHR time series. The authors train autoregressive generative models at multiple temporal resolutions, including a fine-grained event-time baseline and coarser windowed models obtained via Bernoulli-mixture tokenization of within-window presence vectors, and then fuse the experts via a convex combination of horizon-specific predicted probabilities with weights learned on validation data. Across three datasets (MIMIC-IV, a large tertiary-center EHR, and EHRSHOT), seven clinical endpoints, and short/long horizons (30-day and 2-year), they report consistent AUROC and BCE improvements over the single-resolution baseline, with analysis attributing gains to resolution-dependent tradeoffs in discrimination, calibration, and sharpness.

**Compliance With Llm Reviewing Policy:**

Affirmed.

**Final Justification:**

This work is novel and solid. The authors addressed all my concerns during rebuttal. So I would support the acceptance of this paper.

**Key Questions For Authors:**

1. How “zero-shot” is MoRGen in practice if mixture weights are learned per task/horizon on labeled validation data? Can you report results with (a) uniform averaging across experts and (b) a single set of weights shared across tasks/horizons to quantify reliance on supervised fusion?
2. How sensitive are results to the Bernoulli mixture hyperparameters and design (K, EM iterations, k-means++ init)? Do count-augmented or trend-aware window features improve performance on utilization-driven endpoints like readmission?
3. What is the marginal training and inference cost of adding R coarse experts? Please report wall-clock training times per expert, and per-patient rollout latency for baseline vs MoRGen.

**Limitations:**

yes

**Strengths And Weaknesses:**

Strengths

Proposes a simple, stable, and computationally light fusion mechanism (convex mixture over expert probabilities) that avoids per-patient gating and is convex to optimize per task/horizon.

The evaluation spans three independent datasets with multiple outcomes and two horizons, reporting multiple metrics (AUROC, BCE, calibration error, sharpness) and providing task/horizon-dependence analyses that justify the mixture approach by showing no single resolution dominates.

Clear problem setup and metric definitions, with intuitive framing of discrimination/calibration/sharpness tradeoffs as a function of temporal resolution.

Addresses an important pitfall in zero-shot generative forecasting—the mismatch between model time resolution and outcome dynamics—and provides a simple, general recipe to mitigate it.

Weaknesses

The “zero-shot” positioning is weakened by learning task- and horizon-specific mixture weights on labeled validation data; while the experts are zero-shot, the fusion is supervised per task/horizon and may not transfer without labels.

Baselines are limited to single-resolution autoregressive models (fine and coarse); there is no comparison to other prior EHR sequence models listed in Table 1, including ETHOS, AReS, and Event Stream GPT or strong non-generative baselines calibrated for each task.

Limited reporting of computational cost: training 20 coarse models per dataset is nontrivial; inference latency is discussed qualitatively without runtime numbers or cost-quality tradeoffs.

---

> ### Author Rebuttal · Authors · 2026-03-31
>
> # Weaknesses
> 1.	The “zero-shot” positioning is weakened by learning task- and horizon-specific mixture weights on labeled validation data…
>
> Response: Thank you for raising this concern. The individual experts are zero-shot in the sense that they are pretrained generative forecasters and are not fine-tuned for any downstream prediction task. However, the final MoRGen fusion used in the main experiments is not zero-shot, because the convex mixture weights are learned from labeled validation data separately for each dataset, task, and horizon. We will revise the wording throughout to make this distinction explicit.
>
> We additionally evaluated a naïve zero-shot MoE variant and an all task-shared MoE variant. The naïve zero-shot variant is uniform averaging of the expert predictions, this outperforms the baseline in 66.7% of experiments across 3 datasets, 7 tasks, and 2 horizons (30 days and 2 years). Second, shared-weight fusion learns a single set of mixture weights jointly across all tasks and both horizons within each dataset, outperforms the baseline in all experiments we tested. These results suggest that MoRGen’s gains are not solely due to highly task-specific fitting: even a simple averaging-based mixture of experts achieves stronger performance than the baseline.
>
> 2.	Baselines are limited to single-resolution autoregressive models (fine and coarse); there is no comparison to other prior EHR sequence models listed in Table 1, including ETHOS, AReS, and Event Stream GPT or strong non-generative baselines calibrated for each task.
> Response: Please see our response to reviewer gTeZ (4), where we provide a comparison to ETHOS. We also note that results in AReS also use the same underlying ETHOS model.
>
>
> 3.	Limited reporting of computational cost…
> Response: Same as (7) below.
>
> # Key Questions For Authors
> 4.	How “zero-shot” is MoRGen…?
> Response: Same as (1) above.
>
>
> 5.	How sensitive are results to the Bernoulli mixture hyperparameters and design (K, EM iterations, k-means++ init)?
>
>
> Response: We hold the EM procedure fixed across experiments because the chosen initialization and iteration budget were stable in practice. To assess sensitivity to K, we ran single-K MoRGen ablations using the baseline plus coarse experts from one K value per dataset, sweeping four K values per dataset across 3 datasets, 7 tasks, and 2 horizons (168 experiments total). These single-K variants outperform the baseline in 167/168 experiments, indicating only modest sensitivity to the precise choice of K, with the largest variation on EHRSHOT.
>
>
> 6.	Do count-augmented or trend-aware window features improve performance on utilization-driven endpoints like readmission?
>
> Response: Thank you for this thoughtful suggestion. We agree that count-augmented or trend-aware coarse representations are a promising extension of the Bernoulli presence-based compression used here, and such variants may capture complementary temporal structure that is useful for some tasks. At the same time, MoRGen already includes the fine-grained baseline expert, which has access to detailed local ordering and trend information, so the current framework is not limited to coarse presence-based summaries alone. We did not evaluate count- or trend-augmented coarse experts in the current study, so we leave these as an important direction for future work.
>
> 7.	What is the marginal training and inference cost of adding R coarse experts? Please report wall-clock training times per expert, and per-patient rollout latency for baseline vs MoRGen.
>
> Response: Thank you for this important question. We measure compute times on a GH200 machine. The marginal training cost of adding coarse experts can be summarized by the wall-clock time per added expert. On EHRSHOT and MIMIC, the average wall-clock training time for one coarse expert was 13.0 minutes and 39.8 minutes, respectively, including both fitting of the Bernoulli mixture model and autoregressive training. Because the experts are independent, these jobs can be run sequentially on limited hardware or parallelized across GPUs or machines.
> For inference, we measured per-patient rollout latency for baseline vs. MoRGen when generating one trajectory per patient. On EHRSHOT, baseline rollout required 12.83 ms per patient, compared with 270.17 ms for MoRGen when experts were run sequentially and we project it to take 20.39 ms if expert inference were parallelized, which we estimated as the latency of the slowest expert. On MIMIC, baseline rollout required 10.41 ms per patient, compared with 142.58 ms for sequential MoRGen and a projected 12.26 ms for parallel MoRGen. Thus, adding coarse experts increases total inference work, but parallelization substantially reduces wall-clock overhead, yielding 1.6× baseline latency on EHRSHOT and 1.2× on MIMIC. We will add full benchmark details in the appendix and clarify in the paper that MoRGen trades additional compute for improved predictive performance.

---

> > ### Author Rebuttal · Reviewer_aUJc · 2026-04-01
> >
> > I thank the authors for their detailed response, which addresses my concerns. I will maintain my score and continue to support the acceptance of this work.

---

### Official Review · Reviewer_gTeZ · 2026-03-10

**Soundness:** 3
**Presentation:** 2
**Significance:** 3
**Originality:** 3
**Overall Recommendation:** 5
**Confidence:** 4

**Summary:**

In this paper, the authors investigate the performance discrepancies in zero-shot forecasting of EHR data caused by irregular time sampling of the data. They propose both a coarse windows-based approach to fix the temporal resolution , and a mixture of expert model to aggregate models trained at different resolution. They experimentally show that models that does not consider the time resolution struggle to predict adversarial outcomes in the long run; whereas model trained at a fixed resolution display much more interpretable failure points and that when aggregated they provide improved performances.

**Compliance With Llm Reviewing Policy:**

Affirmed.

**Final Justification:**

The authors have provided answers to my questions.

**Key Questions For Authors:**

1/ The construction of the presence vector is only a binary indicator. Other methods are deferred to the Appendix, but this section is missing. In a clinical setting, repeated events in a short period can be a very strong predictor of adversarial outcomes, which will not be correctly represented in the presence vector. Have you consider using something akin to a Dirichlet mixture of multinomials ?
2/ The authors make all of their comparisons with their own baseline. Is this because it basically encompasses the other method described in the related work ? It would be interesting to compare to other SOTA methods that offer forecasting/time-gaps possibilities.
3/ Why are the mixtures weights $w$ shared across patients ? In the LLM literature, such mixtures are usually gated by samples so as to provide adequate weighting of the expert while conserving sparsity. This allows to scale to larger models and increased number of experts with better performances.

**Limitations:**

See weaknesses

**Strengths And Weaknesses:**

Strengths:
The initial problem exposition is clear and pretty intuitive. The time scale of the data used to train model makes it hard to correctly model rare and/or long-ranged adversarial outcome. Following a line of work based on ensembling specialized models trained on specialized task, the authors adequately showcase the pitfalls of the ‘naive’ approach. The coarse windows construction is robust, lightweight, and only introduces a single, easy to tune, hyper-parameter $K$. The extensive experiments on 3 real datasets supported by a coherent set of metrics, provide a good basis to support the initial claim as well as the proposed method. The code used to conduct of the experiment is shared. Albeit the actual code used to train the model is hard to track down.
Weaknesses :
The presentation of the method is very convoluted, more specifically section 4.2. The paragraph “Coarse autoregressive modeling” can be misleading in the way it presents the model’s output. The presentation could benefit from moving the explication on empty windows from the appendix to the main body so as to better explain that the new vocabulary is $\{1,\dots, K\} \cup \{empty_1,\dots,empty_{max}\}$. It could also be beneficial to be explicit that from a forecasted token that is in $\{1,\dots,K\}$ you resample an actual EHR-like token from $\phi_z$ . In the same way maybe explicitly say that the $Y^{(k)}$ are included in the initial vocabulary.
Contrary to the baseline method, the mixture of experts model requires up to R models. This significantly increases the computation budget. Albeit the authors explain that this is mitigated by the fact that the models with large time resolution incur a small amount of output token. Still, in order to scale to much larger models, beyond the 4 layers models used here, this method’s large memory overhead will be problematic. See question 3.

---

> ### Author Rebuttal · Authors · 2026-03-31
>
> # Weaknesses
> 1.	The presentation of the method is very convoluted, more specifically section 4.2…
>
> Response: Thank you for pointing this out. We will revise the main text to clarify that the window vocabulary includes non-empty mixture tokens $z \in \{1,\dots,K\}$ and empty window gap tokens  $\{\Delta_1,\dots, \Delta _{max}\}$, and that non-empty window tokens are decoded into EHR-like tokens. We will also clarify in the appendix how each $Y{(k)}$ is defined for each task in our experiments.
>
>
> 2.	Contrary to the baseline method, the mixture of experts model requires up to R models. This significantly increases the computation budget. Albeit the authors explain that this is mitigated by the fact that the models with large time resolution incur a small amount of output token. Still, in order to scale to much larger models, beyond the 4 layers models used here, this method’s large memory overhead will be problematic. See question 3.
>
> Response: Thank you for this important point. MoRGen does increase total training and inference work relative to a single expert, since it uses multiple resolution-specific experts. However, these experts are independent: they are trained and run separately, do not need to be jointly loaded, and their generated trajectories are cached on disk. The final MoRGen fusion then operates only on these cached expert outputs rather than jointly executing all experts at once. Thus, the pipeline can be run sequentially on limited hardware or parallelized across GPUs/machines when additional resources are available.
>
> # Key Questions For Authors:
> 3.	The construction of the presence vector is only a binary indicator. Other methods are deferred to the Appendix, but this section is missing. In a clinical setting, repeated events in a short period can be a very strong predictor of adversarial outcomes, which will not be correctly represented in the presence vector. Have you consider using something akin to a Dirichlet mixture of multinomials?
>
>
> Response: Thank you for this point. We agree that a binary presence vector is a deliberately lossy within-window representation and does not preserve repeated occurrences or fine local ordering within the same window. Our intent here was not to claim that this is the best possible coarse summary of a window of data, but to study whether a simple temporally coarsened representation can still retain useful and complementary forecasting signal. For many of the long-horizon tasks we study, the clinically relevant signal may depend more on broader trajectory patterns than on precise local ordering within a short window. Empirically, we find that even this simple coarse representation is often competitive and complementary to the fine-grained expert within MoRGen.
>
>
> 4.	The authors make all of their comparisons with their own baseline. Is this because it basically encompasses the other method described in the related work ? It would be interesting to compare to other SOTA methods that offer forecasting/time-gaps possibilities.
>
> Response: Our main goal was to isolate the effect of temporal resolution while holding the forecasting framework fixed. That said, we agree an external comparison is useful, and we add a comparison to ETHOS below (on the same observation vocabulary used by the baseline) as it is a peer-reviewed performant zero-shot autoregressive generation model for EHR data. Note that the AReS paper also uses the same underlying ETHOS model. We train it on our discretized dataset and observe ETHOS achieves higher AUROC than the fine-grained baseline in 73.8% of experiments. MoRGen outperforms ETHOS in 67% of experiments. Since MoRGen is meant to complement any suite of zero-shot experts, we can simply add ETHOS as an additional expert that MoRGen has access to, when we do so MoRGen achieves higher AUROC than ETHOS in 95.2% of experiments.
>
> 5.	Why are the mixtures weights $w$ shared across patients? In the LLM literature, such mixtures are usually gated by samples so as to provide adequate weighting of the expert while conserving sparsity. This allows to scale to larger models and increased number of experts with better performances.
>
> Response: Thank you for this question. We shared the mixture weights across patients because our goal in this work was to test whether coarse experts provide useful complementary signal for a given dataset, task, and horizon using a simple fusion rule. In preliminary experiments on our validation-set fitting setup, patient-specific gated mixture fitting approaches were less stable and did not generalize as well. A gated mixture would be more data hungry and require substantially more expert inference. We therefore chose a shared task- and horizon-specific mixture as a simpler and more robust way to evaluate whether coarse temporal resolutions add predictive value beyond the fine-grained baseline.

---

> > ### Author Rebuttal · Reviewer_gTeZ · 2026-04-03
> >
> > Thanks for the clarification regarding the runtime : caching is a good option to avoid overusing memory. The comparison with ETHOS and its inclusion in the mixture is insightful and deserves to be included in the main paper. Regarding the construction of the coarse window, the suggested approach supports the claim of the paper. Yet I expect the comparison with other variants to be present in the final paper, as *promised* in the paper (see footnote 4!). Just out of curiosity, do you think per patient mixture could be beneficial ? What would it take to make it work ?

---

> > > ### Author Response · Authors · 2026-04-07
> > >
> > > We sincerely thank the reviewer for the thoughtful follow-up, the careful engagement with our rebuttal, and the constructive suggestions for strengthening the final paper. We especially appreciate your positive feedback regarding the runtime clarification, the ETHOS comparison, and the coarse-window construction.
> > >
> > > We also appreciate the reminder about footnote 4. In the final paper, we will include comparisons to additional coarse-window variants, including count-based alternatives, in the appendix, together with individual expert AUROC and BCE.
> > >
> > > We think a per-patient mixture could be beneficial in principle, since different patients may benefit from different temporal resolutions.  However, per-patient mixing would require much more data because it aims to learn patient-specific temporal trends rather than average trends across all patients.  We view this as an interesting direction for future work and we will add a sentence to our discussion section stating this.

---

### Official Review · Reviewer_orTz · 2026-03-11

**Soundness:** 2
**Presentation:** 3
**Significance:** 3
**Originality:** 2
**Overall Recommendation:** 3
**Confidence:** 4

**Summary:**

This paper proposes a multi-resolution-based strategy called MoRGen to more effectively predict clinical events and risks over time in medical EHR data. The proposed method addresses the time window size selection problem encountered in zero-shot prediction using sequence data. Compared to existing studies, it clearly presents distinct problem formulations and methodological differences. It also holds practical value by analyzing not only simple classification performance but also the improvement in prediction probability calibration. Furthermore, extensive experiments were conducted across diverse prediction tasks and time ranges using three large-scale clinical datasets to evaluate the generalizability of the proposed method.

**Compliance With Llm Reviewing Policy:**

Affirmed.

**Final Justification:**

I appreciate the authors' constructive and detailed rebuttal. The clarifications regarding task definitions (lab-related abnormal values) and the planned inclusion of baseline characteristics and comprehensive result tables in the appendix are helpful.

However, I am maintaining my recommendation of Weak Reject. While the authors addressed many of my concerns through clarification, the core issues regarding the comparative evaluation and the "zero-shot" claims remain partially resolved only through promises of future reporting. Specifically, the "zero-shot" framing in the original manuscript was somewhat overstated, and although the authors clarified their definition, the distinction between task-specific MoRGen fusion and true zero-shot performance is critical. Furthermore, while the multi-resolution strategy is interesting, the empirical evidence provided in the current version lacks a unified, transparent comparison (Baseline vs. Coarse vs. MoRGen) that clearly demonstrates the relative gain of each component beyond win rates.

**Key Questions For Authors:**

1. While adjusting the time resolution may be practical, I would like to inquire about your thoughts regarding the potential for information loss.

2. The basic characteristics and detailed information of the population for the data used in the experiment must be added to the appendix.

3. It is difficult to find information on how variables such as the definitions of laboratory-related tasks (leukocyte, platelet, hemoglobin, hematocrit, creatinine) were defined as class labels. Could you provide more detailed information on this?

4. For an objective evaluation, we require not only the results of a multifaceted performance assessment but also detailed experimental information. Can you provide this information?
4-1 a direct comparison between individual Coarse models and MoRGen
4-2 Key metrics, including BCE, should be reported in a unified format comparing Baseline vs. Coarse vs. MoRGen
4-3 Since win rate alone provides limited information, it is necessary to compile the entire experimental results and submit them as an appendix.

5. Questions Regarding the Design of Model Construction and Experimental Evaluation
5-1 Were training and test data separated during model construction?
5-2 Was a pre-trained model constructed that can perform zero-shot learning?
5-3 Was the model evaluated using data with class labels divided into seen and unseen?
5-4 If not, I question whether this constitutes an overstatement of the experiment.

**Limitations:**

The restrictions are stated very clearly.

**Strengths And Weaknesses:**

Strengths

This study is evaluated as rational and practical, as it systematically demonstrated the issues arising from time resolution selection through objective experiments across diverse datasets and proposed methodologies to mitigate them. Furthermore, it clearly describes the difficulties and limitations of applying survival analysis—often discussed alongside sequence models in medical time series forecasting—honestly presenting the scope of its application, which is also positively regarded.

Weaknesses

Need for discussion on information loss due to temporal aggregation.
- Adjusting temporal resolution may be practically beneficial for training stability and computational efficiency. However, temporal aggregation inherently entails potential information loss. The manuscript primarily emphasizes the benefits of coarse resolutions but does not sufficiently discuss what types of information may be lost and how such loss could affect predictive performance or interpretability. A more balanced discussion of this trade-off would strengthen the methodological justification.

Insufficient reporting of baseline characteristics of experimental cohorts.
- The lack of detailed dataset summaries limits reproducibility and interpretability. Providing baseline cohort characteristics—such as the number of patients, event prevalence, follow-up duration, and key variable distributions—would improve transparency and facilitate a better understanding of the experimental settings.

Ambiguity in downstream task definitions.
- Death and readmission tasks are clearly interpretable as binary clinical events. However, it remains unclear what the laboratory-related tasks (e.g., leukocyte, platelet, hemoglobin, hematocrit, creatinine) specifically represent. It should be clarified whether these tasks correspond to laboratory measurement occurrence, abnormal value detection, or other derived clinical labels. Since AUROC is reported, the manuscript should explicitly define how binary labels were constructed for each task.

Lack of transparent comparative evaluation.
- The experimental comparisons are presented in a fragmented manner, separating Baseline vs. Coarse models and Baseline vs. MoRGen, which makes it difficult to assess the relative contribution of each component.(a) Since MoRGen integrates multiple coarse-resolution experts, a direct comparison between individual Coarse models and MoRGen is necessary to understand the benefit of model fusion.(b) Key metrics, including BCE, should be reported in a unified format comparing Baseline vs. Coarse vs. MoRGen to enable clear performance interpretation.(c) Reporting only win rates limits interpretability; providing detailed task- and horizon-specific results in the appendix would substantially improve transparency and result reliability.

Doubts about the zero-shot evaluation setting
- The manuscript describes the model as performing zero-shot forecasting, but it is unclear how the zero-shot setting is defined. In many zero-shot learning frameworks, datasets are typically divided into seen (training) and unseen (test-only) classes to evaluate generalization. However, the current experimental setup appears to use the same tasks for both training and evaluation.

---

> ### Author Rebuttal · Authors · 2026-03-31
>
> # Weaknesses
> >
> 1.	Need for discussion on information loss due to temporal aggregation.
>
> Response: Thank you for this important point. Individual coarse experts can discard fine-grained ordering and timing information within a window. However, MoRGen does not replace the fine-grained baseline; it adds coarse experts on top of it, and the baseline remains one expert in the mixture. Thus, the claim is not that coarse models are universally better, but that they can provide complementary signals to the baseline for some task–horizon pairs.
>
>
> 2.	Insufficient reporting of baseline characteristics of experimental cohorts.
>
> Response: Thank you for pointing this out, we will make the cohort summary more complete. Appendix A already reports the numbers of patients, discharges, and split percentages, and Figure 5 reports task prevalence across post-discharge horizons. We will additionally add an appendix table with median age at discharge, median follow-up duration, and task prevalences at the reported horizons.
>
> 3.	Ambiguity in downstream task definitions.
>
> Response: Thank you for pointing this out. The lab tasks are abnormal-value detection tasks. We will clarify this at the beginning of Section 5: “The tasks evaluated across the three datasets are readmission, death, and abnormal laboratory value detection for leukocyte, platelet, hemoglobin, hematocrit, and creatinine. Specifically, creatinine corresponds to elevated creatinine >2.3mg/dL, while hematocrit, hemoglobin, leukocyte, and platelet correspond to decreased values <24.6%, <8.0 g/dL, <5.1 K/uL, and <86.0 K/uL, respectively. Numeric-valued observations such as labs are discretized into deciles, whereas non-numeric events such as admissions are represented directly as discrete tokens. Accordingly, abnormal-value detection is defined on the discretized laboratory representation by checking whether the value associated with an observed laboratory decile ever exceeds or falls below the corresponding threshold by horizon H.”
>
> 4.	Lack of transparent comparative evaluation.
>
> Response: We agree that this would improve transparency. We will add appendix tables reporting AUROC and BCE for the baseline, MoRGen, and each individual coarse expert for all datasets and both headline horizons (30 days and 2 years). This will allow for direct comparison between the baseline, MoRGen, and the individual coarse experts.
>
> 5.	Doubts about the zero-shot evaluation setting.
>
> Response: Same point as (10) and (11) below: the experts are zero-shot under the generative-pretraining definition used in prior zero-shot EHR forecasting work, but the task-specific MoRGen fusion in the main experiments is not, since its mixture weights are learned from labeled validation data. We will clarify this distinction explicitly.
>
> 6.	While adjusting the time resolution may be practical, I would like to inquire about your thoughts regarding the potential for information loss.
>
> Response: Same point as (1) above; we will clarify this trade-off in the discussion.
>
> 7.	The basic characteristics and detailed information of the population for the data used in the experiment must be added to the appendix.
>
> Response: Same as (2).
>
>
> 8.	It is difficult to find information on how variables such as the definitions of laboratory-related tasks… Could you provide more detailed information on this?
>
> Response: Same as (3).
>
> 9.	For an objective evaluation, we require… detailed experimental information. Can you provide this information?
>
> Response: Same as (4).
>
> 10.	Questions Regarding the Design of Model Construction and Experimental Evaluation
> 5-1 Were training and test data separated during model construction?
>
> Response: 5-1: Yes, we split by patient id into training, validation, and test sets. Experts are trained via next token prediction on the training set and results are reported on the test set. Train/Val/Test split percentages are in Appendix A.
>
> 11.	5-2 Was a pre-trained model constructed that can perform zero-shot learning? 5-3 Was the model evaluated using data with class labels divided into seen and unseen? 5-4 If not, I question whether this constitutes an overstatement of the experiment.
>
> Response: We thank the reviewer for pointing this out. Here, “zero-shot” does not mean a seen/unseen class split; it means the experts are trained only via autoregressive next-token prediction on raw EHR trajectories, not on the downstream forecasting task. Under this definition, both the baseline and coarse experts are zero-shot, with event probabilities estimated from generated futures by Monte Carlo averaging. However, the task-specific MoRGen fusion used in the main experiments is not zero-shot, since its mixture weights are fit on labeled validation data. We will revise the paper to make this distinction explicit. A truly zero-shot variant using uniform averaging still outperforms the baseline in 66.7% of experiments.

---

### Official Review · Reviewer_PKgh · 2026-03-12

**Soundness:** 3
**Presentation:** 3
**Significance:** 2
**Originality:** 2
**Overall Recommendation:** 5
**Confidence:** 2

**Summary:**

The paper presents MoRGEN, a generative framework for handling time series data which exists at multiple timescales. The primary method relies on constructing a series of models over sets of bins of size $\Delta t$ and training an autoregressive model which bins events in intervals of size $\Delta t$. Predictions of future events are generated by sampling the model and applying a weighted average of predictions per event class.

**Compliance With Llm Reviewing Policy:**

Affirmed.

**Final Justification:**

The rebuttal of the paper addresses my main concerns over algorithmic weaknesses by elaborating upon the papers framing. They have convinced me that despite certain algorithmic weaknesses, the overall paper framing makes sense, and is a simple method to address the stated problem. I raise my score to accept as a result, as I believe this improves the overall paper soundness; however, keep my confidence the same, as broader problem statement impact is better gauged by the AC.

**Key Questions For Authors:**

Do the authors propose a method for dealing with incredibly long timescale, where rolling out the full duration of the short length scale model may not be possible (say the short length scale versus the horizon scale varies by many orders)? Have the authors considered mixed timescale / hierarchical approaches to handle these regimes?

Autoregression is known to fail at long context lengths without substantial engineering effort; which is why fine-grained approaches likely fail at longer timescales. Have to authors considered works which are more stable for long generation like diffusion forcing?

Such a comparison may be useful.

https://arxiv.org/abs/2407.01392

**Limitations:**

yes

**Strengths And Weaknesses:**

Pros:
The method presents meaningful improvements over an autoregressive baseline which is “fine grained” level; predicting next events with spacing tokens. (Measured by AUCROC score)
The method ablates the performance of binning sizes, finding “task” specific binning resolutions measured by cross entropy. These ablations are satisfactory in demonstrating that fixed bin size is insufficient.
The method is presented very clearly and motivated clearly. The paper as far as methodology was incredibly easy to read.

Cons:
The model is effective at marginal predictions, but not joint predictions; as the aggregation of a particular event's occurrence is taken by average of many sequences, across many resolutions.
This inflates performance on AUC-ROC. The study and method ablation are limited exclusively to tasks that are computed off of marginal probabilities, which eliminates some benefits which are present within the joint nature of the fine-grained baseline.
While it makes for a better predictor of marginal probabilities, it does not provide a satisfactory way of predicting joint outcomes at multiple timescales.
The overall methodology required rolling out many different binning sizes before aggregating. This seems expensive as it does not generalize to tasks where time scale differs by many orders.

---

> ### Author Rebuttal · Authors · 2026-03-31
>
> # Weaknesses
> 1.	The model is effective at marginal predictions, but not joint predictions; as the aggregation of a particular event's occurrence is taken by average of many sequences, across many resolutions. This inflates performance on AUC-ROC.  The study and method ablation are limited exclusively to tasks that are computed off of marginal probabilities, which eliminates some benefits which are present within the joint nature of the fine-grained baseline. While it makes for a better predictor of marginal probabilities, it does not provide a satisfactory way of predicting joint outcomes at multiple timescales.
>
> Response: Thank you for this insightful comment. We interpret this concern as distinguishing marginal single-event predictions from joint temporal predictions involving multiple events and multiple different horizons. In this paper, we primarily evaluate marginal forecasting tasks, where the goal is to predict whether a code or code set occurs by a given horizon. For example, the MIMIC-IV readmission task is defined as the occurrence of any inpatient admission code within the horizon after discharge. To address the concern of performance on joint tasks, we evaluate MoRGen vs the Baseline model on a suite of additional joint tasks on all three datasets using four endpoint pairs per dataset: creatinine/death, creatinine/platelet, hematocrit/readmission, and leukocyte/hemoglobin or leukocyte/death. For each pair, we fix the first horizon at 30 days and sweep the second horizon over 90, 180, and 730 days, so the label is positive only when both events occur within their respective horizons. In this joint task sweep, the MoRGen achieves higher AUROC on all of the experiments we tested. We will add the full sweep results in the appendix.
>
> 2.	The overall methodology required rolling out many different binning sizes before aggregating. This seems expensive as it does not generalize to tasks where time scale differs by many orders.
>
> Response: Thank you for this important point. MoRGen does increase total training and inference work relative to a single expert, since it uses multiple resolution-specific experts. However, these experts are independent: they are trained and run separately, their generated trajectories are cached, and the final MoE fusion operates only on these cached outputs rather than jointly executing all experts. Thus, the pipeline can be run sequentially on limited hardware or parallelized across GPUs/machines. Our goal is not to claim that a fixed set of resolutions will universally cover tasks spanning arbitrarily many orders of magnitude in timescale, but to show that combining a small number of resolutions is effective in our setting. The joint-task results in (1) and longer-horizon results in (3) support that MoRGen remains effective across substantially different queried horizons using this fixed set of resolutions.
>
>
> 3.	Do the authors propose a method for dealing with incredibly long timescale, where rolling out the full duration of the short length scale model may not be possible (say the short length scale versus the horizon scale varies by many orders)?
>
> Response: Thank you for this question. In the main paper, we capped the horizon sweep at 2 years as part of our predefined evaluation setup across datasets and tasks. To directly address whether the gains extend further, we additionally evaluated substantially longer horizons of 5, 10, and 20 years. MoRGen continues to outperform the baseline in AUROC at all tested longer horizons. Across the 3 datasets, 7 tasks, and 3 added horizons, MoRGen improves over the baseline in all experiments we tested. We will add the full tables of results to the appendix.
>
> 4.	Have the authors considered mixed timescale / hierarchical approaches to handle these regimes?
>
> Response: Thank you for this question. Yes, in early iterations of this work, we explored a mixed-timescale/hierarchical variant that interleaved coarse window-level tokens with fine-grained observation tokens in a single sequence. The autoregressive mixed timescale model would have to first generate a coarse window-level token before generating the fine-grained  observation tokens. In our preliminary experiments, this approach did not improve performance reliably relative to the baseline model, so we did not pursue it further in the main paper. We will summarize these preliminary results in the appendix.
>
>
>
> 5.	Autoregression is known to fail at long context lengths without substantial engineering effort; which is why fine-grained approaches likely fail at longer timescales. Have to authors considered works which are more stable for long generation like diffusion forcing?
>
> Response:
>
> We thank the reviewer for this suggestion. Methods aimed at improving long-sequence generation stability, including state-space models and diffusion-forcing-style approaches, are a promising direction for future work. We will add a sentence to the limitations and future work section highlighting this.

---

> > ### Author Rebuttal · Reviewer_PKgh · 2026-04-03
> >
> > I thank the authors for their detailed response; they resolve a decent portion of concerns and I raise my score to an accept.

---

### Decision · Program_Chairs · 2026-04-30

**Decision:**

Accept (regular)

**Comment:**

The authors propose to build a mixture of autoregressive EHR generative experts trained at multiple temporal resolutions to improve forecasting across tasks/horizons whose dynamics are mismatched to any single discretization.

While the reviewers praise the motivation as intuitive and important, liked the extensive multi-dataset evaluation, and thought the convex fusion approach was simple and practical, they also raise several concerns, including that the “zero-shot” framing was overstated because fusion weights are learned with labels, compute cost is higher than a single expert, and the original comparison tables/baselines were not transparent enough.

The authors' rebuttals clarified the distinction between zero-shot experts and supervised fusion, added joint-task and much longer-horizon results, promised unified appendix tables comparing baseline/coarse/MoRGen, added ETHOS comparisons, and provided runtime measurements plus shared-weight/uniform-weight variants.
The reviewers indicated that these responses have addressed most of their criticisms.

Therefore, we have decided to accept the paper for presentation at ICML. We would still recommend that the authors take the reviewers' feedback into account when preparing the camera-ready version.